



# A multi-year global methane data set obtained by merging observations from TROPOMI and IASI

Kanwal Shahzadi[1,2], Matthias Schneider[1], Nga Ying Lo[1,2], Frank Hase[1], Jörg Meyer[2], Ugur Cayoglu[2], Tobias Borsdorff[3], and Mari C. Martinez-Velarte[3]

[1]Institute for Meteorology and Climate Research (IMKASF), Karlsruhe Institute of Technology (KIT), Karlsruhe, Germany
[2]Karlsruhe Institute of Technology, Scientific Computing Center
[3]Space Research Organisation Netherlands (SRON), Leiden, Netherlands

**Correspondence:** Kanwal Shahzadi (kanwal.shahzadi@kit.edu) and Matthias Schneider (matthias.schneider@kit.edu)

**Abstract.** Data products of atmospheric methane ($CH_4$) with improved vertical sensitivity in the lower troposphere are crucial for gaining a more comprehensive understanding of the impact of anthropogenic emissions. This study presents a $CH_4$ data product derived from the synergetic combination of level 2 (L2) data from TROPOMI (Tropospheric Monitoring Instrument) and IASI (Infrared Atmospheric Sounding Interferometer), specifically $CH_4$ total column and $CH_4$ profiles, respectively. IASI enables high-quality observation of $CH_4$ mixing ratios in the upper troposphere and lower stratosphere, and TROPOMI observations excel in providing sensitivity to the total column-averaged mixing ratio of $CH_4$. By combining the IASI and TROPOMI L2 products synergetically, we can detect tropospheric $CH_4$ (mixing ratios averaged over a layer from the surface up to 450 hPa) that is not significantly affected by the strong $CH_4$ variations around the tropopause. This is not achievable by using IASI or TROPOMI data alone.

For the synergetic L2 data combination, we use the method as presented in detail in Schneider et al. (2022b), and apply it to combine about 444 million individual and high-quality TROPOMI observations with about 805 million individual and high-quality IASI observations made globally over 42 months (from January 2018 to June 2021). The combination method is fast; it uses a tool designed for efficient geo-matching between large data sets and a computationally cheap Kalman filter for calculations and for merging the data sets. We show that the combined data set has a good global coverage. Moreover, we document that the sensitivity (response of the combined data product to real atmospheric $CH_4$ variations) is extremely satisfactory throughout the globe, and the uncertainties are generally below 12-15 ppbv. Furthermore, we demonstrate the increased scientific value of the combined data product when compared to the two individual data products.

The data set of the combined product consists of about 289 million individual data points, and it is provided as NetCDF files. One file has a typical size of 280 MB and contains all data for observations made in one day (the universal time of the TROPOMI observations are taken as the reference time). For review, the data are accessible at https://radar.kit.edu/radar/en/dataset/wq583rnzpmd83m5g?token=UEuECSWHlGgwWBdoPVsvI (Shahzadi et al., 2025) and made freely available at https://www.imk-asf.kit.edu/english/CH4-synergy-IASI-TROPOMI_RemoTeC.php.





## 1 Introduction

The abundance of remote sensing data offers a valuable opportunity to explore and exploit the diversity of data sets. One can
increase the scientific impact of such data by integrating them and by leveraging their different characteristics (like different
sensitivities or error patterns). The integration of diverse data sets can lead to strong synergies, i.e. a new data set having
advanced sensitivities and/or reduced errors.

For an optimal synergetic combination of two data sets, the precise knowledge of the characteristics of the two individual
data sets is essential. Concerning remote sensing data, we can only successfully combine the data sets, if each data point
is made available together with its error (or error covariances), sensitivity/representativeness (i.e. the averaging kernels) and
information on a priori choices.

The vertical sensitivity of satellite remote sensing data for trace gas depends heavily on the spectral region of the observation.
For instance, the Infrared Atmospheric Sounding Interferometer (IASI) aboard the Metop satellites measures nadir spectra in
the thermal infrared region and has a high spatial resolution with global coverage twice daily. IASI has a good sensitivity of
trace gas variations in the free troposphere and the lower stratosphere (e.g. Clerbaux et al., 2009). However, it lacks sensitivity
in the lower troposphere due to low thermal contrast near surface.

Data generated from observations of the Tropospheric Monitoring Instrument (TROPOMI) aboard the Sentinel-5 Precursor
satellite are promising for complementing this deficit in the IASI data. TROPOMI offers a similar good spatial resolution and
coverage as IASI, but it observes Earth surface reflected solar spectra in the near infrared region (e.g. Veefkind et al., 2012).
TROPOMI offers good sensitivity throughout the whole atmosphere and provides total column-averaged trace gas products at
a good quality.

Schneider et al. (2022b) shows that the well-characterised IASI and TROPOMI methane ($CH_4$) data products can be suc-
cessfully combined, and the combined product is superior to the individual data products. The combination retains the good
data quality of $CH_4$ in the free troposphere and lower stratosphere (offered by the IASI product), and of the $CH_4$ total column
(offered by the TROPOMI data). In addition, it yields good-quality tropospheric $CH_4$ data which is not observable in the two
individual data products.

In this study, we present a data set generated with the Schneider et al. (2022b) method using IASI and TROPOMI observa-
tions from January 2018 to June 2021. In Sect. 2, we present the two satellite data sets used for generating a synergetic product,
and discuss their main characteristics and coverages. Section 3 briefly describes the method used for the synergetic data set
combination and presents the achieved data coverage. Section 4 documents the vertical representativeness of the TROPOMI,
IASI and the combined data products, therby revealing the synergetic gain achieved by the data set combination. Moreover,
the global sensitivity patterns of the combined data products are also discussed. In Sect. 5, we present the global pattern of
the leading error sources and Sect. 6 compares the temporal and spatial patterns of the combined data products, demonstrating
their additional scientific potential on top of the individual TROPOMI and IASI data products. Section 8 gives a summary and
outlook.





## 2   Input data products

### 2.1   MUSICA IASI

We use a data product obtained from measurements of IASI, a remote sensing instrument on board of the Metop series of satellites. IASI detects the infrared radiations that are emitted by the Earth and transported throughout the atmosphere. We use the IASI $CH_4$ L2 data product generated by the retrieval processor MUSICA (García et al., 2018; Schneider et al., 2022a) developed at the Karlsruhe Institute of Technology (KIT). MUSICA stands for "MUlti-platform remote Sensing of Isotopologues for investigating the Cycle of Atmospheric water" (project phase 2011 - 2016) which is intialised and developed under the framework of the European Research Council. The MUSICA IASI full retrieval data product versions 3.2 and 3.3, used here, includes trace gas profiles of $H_2O$, the $HDO/H_2O$ ratio, $N_2O$, $CH_4$ , and $HNO_3$. The data are provided together with detailed information on a priori usage, constraint, averaging kernels and error covariances for each individual observation. The footprint pixel at nadir has a diameter of $12\,km$ and IASI offers daily global coverage over both the land and ocean twice. Figure 1 shows the coverage for December, January and February (DJF) in (a) and for June, July and August (JJA) in (c). Some data gaps are observed over dry stony/sandy regions, like the Sahara desert. There are significant number of retrieval results filtered out due to poor quality of the respective spectral fits. The reason for this poorer fit quality is the weak and poorly-decribed infrared surface emissivity of stony/sandy ground (e.g. Zhou et al., 2011). MUSICA IASI version 3.2 and 3.3 data are described in detail in Schneider et al. (2022a) and made freely available for October 2014 to July 2021 (Schneider et al., 2021). Here, we work with the period from January 2018 - June 2021, which is the time when TROPOMI data are available along with IASI's.

We only use the data if the fit quality of the MUSICA IASI retrieval is very good (`musica_fit_quality_value`) is 3, representing high-quality fits for which the spectral residuals are close to the instrumental noise (see Sect. 6 of Schneider et al., 2022a). Furthermore, we require that the EUMETSAT L2 cloudiness assessment summary flag is 1 (IASI field of view is clear) or 2 (IASI field of view is processed as cloud-free but small cloud contamination is possible). In the latter case, we require additionally an EUMETSAT L2 fractional cloud cover value of 0.0 or NaN. The total amount of individual IASI $CH_4$ L2 data used in this study from the January 2018 - June 2021 period is aproximately 805 million observations. Table 1 gives an overview on the data volumes.

### 2.2   RemoTeC TROPOMI

The second data set is the TROPOMI product generated by the retrieval algorithm RemoTeC (Butz et al., 2011; Lorente et al., 2021) at the Space Research Organisation Netherlands (SRON). The TROPOMI data processing was carried out with the Dutch National e-infrastructure with the support of the SURF cooperative. TROPOMI is aboard the Sentinel-5 Precursor (S5P) satellite and provides data since 2018. For this study, we use the beta version of the operational S5P product (Lorente et al., 2023), which uses an updated fit of the surface reflectance spectral dependency to a third-order polynomial fit. The data are made freely available at https://ftp.sron.nl/open-access-data-2/TROPOMI/tropomi/ch4/19_446/.

The mean data coverage for January 2018 - June 2021, particularly seasons DJF and JJA is shown in Fig. 1(b, d). TROPOMI offers a near-global coverage over land at the resolution of $7\,km \times 7\,km$ since its launch in October 2017 (upgraded to $5.5\,km \times 7\,km$





in August 2019). Over ocean, observations are only possible in glint mode, which leads to much less frequent observations.
Its coverage also differs seasonally. For example, no data are available at high latitudes over winter hemispheres, because high solar zenith angles are filtered out and TROPOMI cannot observe during polar night. Data coverage over land is best in the subtropics and worst in the tropics, which can be explained by a strict cloud filtering. The TROPOMI $CH_4$ L2 data is available with the column averaging kernels, errors and an a priori $CH_4$ profile data. These a priori data are calculated by the global chemistry transport model TM5 (Krol et al., 2005). We use TM5 a priori data as the common TROPOMI and IASI $CH_4$ a a priori data, i.e. we adjust the IASI product to the TROPOMI a priori data (Rodgers, 2000). In order to exclude data of a compromised quality, we only use the TROPOMI data that have a quality flag value (`qa_value`) of 1.0. In this study we use about 444 million TROPOMI observations made between January 2018 and June 2021 (for more details on the data volumes see Table 1).

**Table 1.** Table summarizing data volumes in million. The values are given for the whole 42 months (All) and separately for the different seasons. DJF: December, January, February; MAM: March, April, May; JJA: June, July, August; SON: September, October, November

| Season | IASI | TROPOMI | Combined |
|---|---|---|---|
| Total observations | 805 | 444 | 289 |
| December, January, and February (DJF) | 220 | 126 | 84 |
| March, April,and May (MAM) | 225 | 95 | 75 |
| June, July, and August (JJA) | 177 | 100 | 65 |
| September, October, and November (SON) | 183 | 123 | 66 |

## 3 Synergetic data product

In this section, we briefly describe the method used for combining the IASI and TROPOMI L2 data, and present the data amount and coverages of the combined data product. We use the validated method as described in Schneider et al. (2022b). So far, the method has only been applied for creating small data sets (representing individual months or limited areas). Here we apply the method to create a much more substantial data set.

### 3.1 Geomatching

In order to combine the data, we have to identify the IASI observation that best matches in time and space with a given TROPOMI observation, i.e. we have to optimally geomatch the data sets. This is a significant challenge for large data sets and we developed a dedicated geomatching algorithm based on the work of Ameri et al. (2014). The geomatching requires metadata information such as spatial, temporal and sensor-specific attributes of the individual sensors, which are stored in a MongoDB database. We chose MongoDB due to its flexible handling of semi-structured JSON-like data, native support for spatial queries,





and horizontal scalability. These features enable a fast selection of daily observations and a simplified geospatial filtering during the initial matching phase, depending on the distribution strategy employed and the availability of computational resources. The metadata are loaded as in-memory data frames using Python's Pandas library (McKinney, 2010) for further processing. Afterwards, vectorised-distance calculations, using haversine distance, was implemented to identify IASI observations near each TROPOMI point within the defined temporal window. In a first geomatching step, we use all IASI observations, within 50 km and 6 h of TROPOMI observation. Furthermore, we require that the surface pressure difference of both observations is within 50 hPa. These requirements are generally fulfilled for many IASI observations. In a second geomatching step, we select the optimal match as the minimum of the Euclidean distance calculated from normalized horizontal distances, time differences and surface pressure differences. The normalization for the horizontal distance is 50 km, for the temporal distance it is 2 hours, and for the surface pressure difference is 5 hPa.

Figures 2(a, c) and 2(b, d) show the mean spatial and temporal mismatches, respectively, obtained for the optimal matches between TROPOMI and IASI for DJF and JJA. The spatial mismatches are mostly below 30 km, except for regions where IASI data are sparse (e.g. over the Sahara). The temporal mismatches show a clear latitudinal gradient. The temporal mismatches are smallest in middle and high northern latitudes (less than 2 hours) and largest in middle southern latitudes (generally more than 4 hours). This can be explained by the different orbits of the two satellites that carry the IASI and the TROPOMI instruments.

## 3.2 Combination by a Kalman filter

The Schneider et al. (2022b) method uses a Kalman filter to optimally combine the IASI and TROPOMI data. Assuming moderately non-linear IASI and TROPOMI retrieval processes, it has been shown that the method is analogous to performing a retrieval that simultaneously uses the level 1 spectra of TROPOMI and IASI. However, it remains computationally inexpensive and automatically benefit from the most latest improvements made by the individual IASI and TROPOMI retrieval experts.

The Kalman filter approach is analogous to a data assimilation approach, where a background state $\boldsymbol{x}^{\mathrm{b}}$ is improved by adding information provided by a measurements $\boldsymbol{y}$. The result is the analysed state $\boldsymbol{x}^{\mathrm{a}}$:

$$\boldsymbol{x}^{\mathrm{a}} = \boldsymbol{x}^{\mathrm{b}} + \mathbf{G}[\boldsymbol{y} - \mathbf{H}\boldsymbol{x}^{\mathrm{b}}], \tag{1}$$

where $\mathbf{H}$ is the measurement operator that projects the background state onto the measurement domain, and $\mathbf{G}$ is a Kalman gain matrix, that describes how inconsistencies between the background state ($\boldsymbol{x}^{\mathrm{b}}$) and the measurement ($\boldsymbol{y}$) impact on the analysis state ($\boldsymbol{x}^{\mathrm{a}}$).

$$\mathbf{G} = \mathbf{S}^{\mathbf{b}}\mathbf{H}^{T}\left[\mathbf{H}\mathbf{S}^{\mathbf{b}}\mathbf{H}^{T} + \mathbf{S}_{\epsilon}\right]^{-1} \tag{2}$$

Here $\mathbf{S}^{\mathbf{b}}$ captures the background covariances and $\mathbf{S}_{\epsilon}$ are the measurement error covariances.

We apply the Kalman filter as described in Eqs. (1) and (2) to the IASI and TROPOMI $CH_4$ L2 data. As mentioned in Sect. 2, we use an IASI profile retrieval result obtained by applying the same $CH_4$ a priori information as in the TROPOMI




retrieval (i.e. the TM5 model calculations). The difference between the retrieved IASI profile and this a priori profile is used
as the background state (i.e. for $x^b$ we use $(\hat{x}_I - x_a)$ where $\hat{x}_I$ is the retrieved IASI CH$_4$ profile and $x_a$ the a priori profile).
The difference between the retrieved TROPOMI XCH$_4$ and the a priori XCH$_4$ is used as the measurement (i.e. for $y$ we use
$(\hat{x}_T^* - x_a^*)$, where $\hat{x}_T^*$ is the retrieved TROPOMI XCH$_4$ value and $\hat{x}_a^*$ is the apriori XCH$_4$ value). The obtained analysis state is
then the difference between the combined profile and the apriori CH$_4$ profile (i.e. $x^a$ is replaced by $\hat{x} - x_a$). The TROPOMI
averaging kernels are used as the measurement operator (i.e. for $\mathbf{H}$ we use $a_T^{*T}$). With these substitutions Eq. (1) is written as:

$$\hat{x} - x_a = (\hat{x}_I - x_a) + g[(\hat{x}_T^* - x_a^*) - a_T^{*T}(\hat{x}_I - x_a)]. \tag{3}$$

The subindices "$I$" and "$T$" stand for IASI and TROPOMI, respectively. The variables with no capital letter subindex stand for
the combined product. The subindex "$a$" indicates the TM5 a priori data and the superindex "$*$" variables that represent total
column-averaged mixing ratio data. Retrieved data are identified by the hat symbol "$\hat{}$". The Kalman gain operator $g$ is written
as:

$$g = \mathbf{S_I}a_T^* \left(a_T^{*T}\mathbf{S_I}a_T^* + S_{T,n}^*\right)^{-1}, \tag{4}$$

which is obtained by substituting, in Eq. (2), the background covariances ($\mathbf{S^b}$) by the IASI a posteriori covariances ($\mathbf{S_I}$) and
the measurement error covariance ($\mathbf{S_\epsilon}$) by the TROPOMI error variance ($S_{T,n}^*$). Equations (3) and (4) are analogous to Eqs. (1)
and (2) of Schneider et al. (2022b). For brevity, we omit the transformation from a linear to a logarithmic scale and assume all
variables are given in linear scale.

The TROPOMI data are provided as total column-averaged mixing ratios (XCH$_4$). From the IASI and the combined CH$_4$
profiles, we calculate three different partial column products: the total column averaged mixing ratio (XCH$_4$), the tropospheric
column-averaged mixing ratio (troXCH$_4$, surface to 450hPa), and the upper tropospheric and stratospheric column-averaged
mixing ratio (utsXCH$_4$, 450 hPa to top of atmosphere). The column-averaged mixing ratio CH$_4$ products refer to the integrated
amount of CH$_4$ integrated over a specific atmospheric layer relative to the amount of dry air in that layer, i.e. it is a dry air mole
fraction of CH$_4$ and it is given in ppbv.

This calculations are made according to

$$\hat{x}^* = w^{*T}\hat{x}, \tag{5}$$

where the pressure weighted resampling operator $w^{*T}$ is a vector of $(1 \times n)$ dimension that resamples the mixing ratio profiles
represented in $n$ vertical levels onto an averaged mixing ratio value and is representative for a certain atmospheric partial
column layer. This operator $\mathbf{w}^{*T}$ is obtained by

$$w^{*T} = (w^T\mathbf{Z}w)^{-1}w^T\mathbf{Z}, \tag{6}$$

where the vector $w$ integrates the targeted partial column layer. It has the $(n \times 1)$ dimensions, with elements being 1.0 for the
levels belonging to the targeted partial column layer and 0.0 for the levels outside of the targeted column layer. For converting





mixing ratio profiles into amount profiles and vice versa, we need the pressure weighting operator $\mathbf{Z}$. It is a diagonal matrix whose elements report the amount of dry air molecules represented by the corresponding vertical level (for more details see Appendix D2 in Schneider et al., 2022b).

### 3.3   Data amount and coverage

From the geomatching, we identify about 289 million TROPOMI observations, for which we have a collocated IASI obser-

vation (for more details on the data volume see Table 1). The respective data coverage for DJF and JJA is shown in Fig. 3. Generally, over land there are more than 5-10 observations per day in a $50\,\text{km}\times50\,\text{km}$ box, except for high latitudes in the winter hemisphere and for the tropics. This coverage is similar to maps showing the TROPOMI data (Fig. 3(a-b), i.e. the coverage of the synergetic data product is strongly determined by the data availability of TROPOMI). Generally, If the TROPOMI data set provides a high-quality data, there is also a closeby high-quality IASI observation available.

## 180   4   Vertical representativeness and sensitivity

Atmospheric trace gas remote sensing data products do not equally represent all the vertical ranges equally (limited vertical representativeness) and they do not always respond to the full amplitude of the real atmospheric trace gas variations (limited sensitivity). This inherent characteristics of trace gas remote sensing products is captured by the remote sensing averaging kernels. Thus, the averaging kernels are indispensable for a correct remote sensing data usage. In this section, we discuss the

averaging kernels of the TROPOMI and the IASI data products, and compare it to the respective kernels of the combined data product. Moreover, we document the dependency of the vertical representativeness and sensitivity on latitude and surface elevation.

### 4.1   Averaging kernels

The TROPOMI data are distributed as a total column-averaged data product ($XCH_4$) together with a total column amount

averaging kernel, which is a row vector $\boldsymbol{a}^T$ of $1 \times n$ dimension ($n$ is the number of considered atmospheric pressure levels). This row vector describes how the retrieved total column amount responds to changes of trace gas amounts at a certain pressure levels (by how many CH4 molecules does the retrieved total total column amount changes when adding one CH4 molecule at a certain pressure level).

     The IASI data are distributed as a vertical mixing ratio profile data product together along with its vertical mixing ratio

averaging kernels, i.e. for each retrieval level there is a dedicated averaging kernel that accounts for how a change in mixing ratio in the real atmosphere affects the retrieved mixing ratio profile. This averaging kernel is a matrix $\mathbf{A}$ of dimension $n \times n$.

     The data combination calculations, as described in Sect. 3.2, also generate a vertical mixing ratio profile. The mixing ratio averaging kernel matrix of $n \times n$ dimension can be calculated by (see also Eq. (3) of Schneider et al., 2022b):

$$\mathbf{A} = \mathbf{A_I} + m\boldsymbol{a_T^*}^T(\mathbf{I} - \mathbf{A_I}). \tag{7}$$





Here, $\mathbf{A}$ and $\mathbf{A_I}$ represent the mixing ratio averaging kernel for the combined data and the IASI product, respectively. The row vector $\boldsymbol{a_T^*}^T$ is the total column-averaged mixing ratio averaging kernel of the TROPOMI product, which can be calculated from the respective total column amount averaging kernel (the row vector $\boldsymbol{a_T}^T$) as follows (see also Appendix D in Schneider et al., 2022b):

$$\boldsymbol{a_T^*}^T = (\boldsymbol{w}^T \mathbf{Z} \boldsymbol{w})^{-1} \boldsymbol{a_T}^T \mathbf{Z}. \tag{8}$$

Here $\boldsymbol{w}^T$ is a row vector that integrates the whole column, i.e. it has dimension $1 \times n$ and all elements have the value 1.0.

## 4.2 Vertical representativeness

Here, we document the vertical representativeness of the total and partial column-averaged data products by means of the total and partial column amount averaging kernels. The TROPOMI data set directly provides the total column amount averaging kernels. For the IASI and the combined data products, we calculate the column amount averaging kernels ($\boldsymbol{a}^T$) from the

provided mixing ratio averaging kernels ($\mathbf{A}$) according to

$$\boldsymbol{a}^T = \boldsymbol{w}^T \mathbf{Z} \mathbf{A} \mathbf{Z}^{-1}. \tag{9}$$

Here, $\boldsymbol{w}^T$ is a row vector of dimension $1 \times n$, whose elements are 1.0 for the levels belonging to the targeted partial column layer and 0.0 for the levels outside of the targeted column layer.

In this subsection, we discuss the total column and partial column amount averaging kernels ($\boldsymbol{a}^T$) of the different data prod-

ucts. Figure 4 shows typical averaging kernels for the individual IASI and TROPOMI products and for the combined product. The black horizontal line indicate the $450 \, \text{hPa}$ level, which we use for separating the different partial columns. We define the atmosphere above $450 \, \text{hPa}$ as the troposphere and the atmosphere below $450 \, \text{hPa}$ as the upper troposphere/stratosphere. Figure 4(a) depicts the total column averaging kernels of TROPOMI (grey line and symbols). It is close to 1.0, indicating the good sensitivity throughout the atmosphere.

In Fig. 4(b), the IASI averaging kernels are shown (grey for the total column, blue for the upper troposphere/stratosphere, and red for the troposphere). The IASI XCH$_4$ product is mainly sensitive to methane between 700 and $100 \, \text{hPa}$ (total column averaging kernel values close to 1.0, blue colour). IASI has a rather limited sensitivity above $700 \, \text{hPa}$ and the sensitivity is rather poor close to the surface (total column averaging kernel values below 0.1, grey colour). IASI is well-suited for measuring the upper tropospheric/stratospheric methane concentrations. The respective averaging kernel (blue colour) has values close to

1.0 for all altitudes between 400 and $100 \, \text{hPa}$. The missing sensitivity of IASI for surface-near methane is illustrated by the tropospheric averaging kernel (red colour), whose values are below 0.5 for all pressure levels above $700 \, \text{hPa}$.

The averaging kernels of the combined product are depicted in Figure 4(c). The total column averaging kernels of the combined product (grey colour) and the individual TROPOMI product are very similar, indicating that almost all information needed for detecting XCH$_4$ is provided by TROPOMI, and IASI only contribute weakly with additional information. Concern-

ing the upper troposphere/stratosphere, the averaging kernel of the combined product (blue colour) is very similar to the IASI averaging kernel, indicating in turn that the information for detecting utsXCH$_4$ comes from IASI and TROPOMI adds only





a very small amount of additional information. The partial column kernel representing the combined troXCH$_4$ product (red colour) is significantly different from the respective IASI averaging kernel. It has high values (between 0.8 and 1.0) from the surface up to about $600\,\mathrm{hPa}$ and low values for the upper troposphere/stratosphere (for pressures below $450\,\mathrm{hPa}$ the values are

$\leq 0.5$). This documents that the combined troXCH$_4$ data product does well detect the CH$_4$ variations in the surface near troposphere and furthermore, it is not significantly affected by CH$_4$ variations that may occur in the upper troposphere/stratosphere. The combined data product is superior to the individual TROPOMI and IASI data products: while the XCH$_4$ and utsXCH$_4$ data of the combined product are representative for total column-averages and partial upper tropospheric/stratospheric column averages as the respective TROPOMI and IASI data products.And its only the combined product offers useful troXCH$_4$ data

(TROPOMI or IASI alone cannot detect the lower tropospheric CH$_4$).

In the sections and figures of the remainder of this paper, we focus on the discussion of the data characteristics of the combined products. For XCH$_4$, the data characteristics are similar to the respective TROPOMI product, for utsXCH$_4$ they are similar to the respective IASI product, and for troXCH$_4$ they are exclusive for the combined product.

### 4.3   Global sensitivity patterns

This section gives some insight into the global variability of the sensitivities. As a measure for the sensitivity, we use the vertical profile averaging kernel matrix and calculate the sum of the diagonal elements representing the vertical levels of XCH$_4$ (all vertical levels, i.e. we calculate the trace of the full matrix), the vertical levels of utsXCH$_4$ (levels below $450\,\mathrm{hPa}$) and the vertical levels of troXCH$_4$ (levels above $450\,\mathrm{hPa}$). In the following, we refer to these sums of the diagonal elements of averaging kernels as the degree of freedom for signal (DOFS).

Figure 5 shows global maps of $50\,\mathrm{km}\times50\,\mathrm{km}$ averages of DOFS values for DJF (first row, Fig. 5(a, c)) and for JJA (second row, Fig. 5(b, d)). The first column of panels (Fig. 5(a, d)) shows the DOFS values for the XCH$_4$ data product. Please note that with calculations according to Eq. (3), we get a combined profile product and the respective mixing ratio profile averaging kernels can be calculated according to Eq. (7). From this mixing ratio profile averaging kernel, we calculate the DOFS of the combined XCH$_4$ product. It is is typically 2-3 and shows a latitudinal dependency. It is largest at low latitudes and lowest at

middle/high latitudes of the winter hemisphere.

The second column of panels (Fig. 5(b, e)) depicts the averaged DOFS values for the utsXCH$_4$ data product. The DOFS values are typically between 1.2 and 2.0, and there is also a clear latitudinal dependence. As for the XCH$_4$ data product, we observe highest values at low latitudes and lowest values at high latitudes of the winter hemisphere.

The latudinal gradient in the XCH$_4$ and utsXCH$_4$ DOFS values are due to a respective latitudinal gradient in the tropopause

pressure level, which is about $400\,\mathrm{hPa}$ at high latitudes and $100\,\mathrm{hPa}$ at low latitudes. Above the tropopause CH$_4$ concentrations decrease sharply. A high tropopause pressure — as encountered at high latitudes — leads to low total amounts of CH$_4$. This in turn results in weak spectroscopic CH$_4$ signatures and thus lower DOFS values for the total atmospheric layer and the upper tropospheric/stratospheric layer. The variation of the tropospheric pressure does not affect the tropospheric layer we observe no latitudinal gradient in Fig. 5(c,f), because we define this layer to be above $450\,\mathrm{hPa}$, i.e. at pressure levels that are higher than

the highest tropopause pressure. Figure 6 visualises the dependence of the DOFS values on the tropopause pressure.





Moreover, in Fig. 5(b, e), we observe spots of high utsXCH$_4$ DOFS in line with high surface elevation, i.e. for the Himalaya, inner Greenland or Antarctica, we observe much higher DOFS values than nearby areas with lower surface elevations. This apparent dependence on the surface elevation is even more pronounced in the troXCH$_4$ DOFS values (Fig. 5(c, f)), but reversed if compared to the utsXCH$_4$ DOFS. Meanwhile for observations over the ocean or over low level land, the troXCH$_4$ DOFS

values are generally between 1.0 and 1.3. For high surface elevation, (Himalaya, Rocky Mountains, Andes, inner Greenland, Antarctica) the DOFS values are often below 1.0.

The dependence of the DOFS values on surface elevation is visualised in Fig. 6. For XCH$_4$, we observe almost no dependence, for utsXCH$_4$ low surface pressure makes high DOFS values more likely, and for troXCH$_4$ there is a clear correlation between surface pressure and DOFS (for a surface pressure of 700 hPa and 1000 hPa the DOFS values are typically 0.75 and

1.0, respectively). This strong dependency comes from our separation of the total atmosphere into two layers: the tropospheric layer and upper tropospheric/stratospheric layer. We define the atmosphere above 450 hPa as tropospheric layer. For low surface pressures this layer has a smaller depth and is consequently represented by a smaller number of levels than for high surface pressures. This results in lower troXCH$_4$ DOFS values for observations over elevated areas, i.e. over areas with low surface pressures. We define the atmosphere below 450 hPa as tropospheric/stratospheric layer. The sum of the DOFS for the upper

tropospheric/stratospheric layer and the tropospheric layer equals the DOFS of total atmospheric column. If the DOFS of the total atmospheric layer does not change strongly with the surface elevelation, a lower DOFS value for the tropospheric layer related to a high surface elevation must come along with a higher DOFS value of the upper troposphere/stratosphere layer. Consequently, the DOFS values of utsXCH$_4$ are systematically larger for surface pressures below 800 hPa compared to surface pressures above 800 hPa.

## 5 Errors

In this section, we analyse the errors of the combined data products. We consider two kind of errors: the noise error caused by the measurement noise of the sensors and deficits in correctly modelling the spectra (e.g. due to insufficient knowledge of surface emissivity or albedo and spectroscopic line shapes or line intensities), and the dislocation error caused by spatial and temporal mismatches.

### 5.1 Noise error

According to Eq. (5) of Schneider et al. (2022b), the noise error covariances of the combined data product can be calculated from the noise errors of the TROPOMI data product (the variance $S^*_{T,n}$) and from the IASI noise error covariances ($\mathbf{S_{I,n}}$):

$$\mathbf{S_n} = (\mathbf{I} - \boldsymbol{g}\boldsymbol{a_T^*}^T)\mathbf{S_{I,n}}(\mathbf{I} - \boldsymbol{g}\boldsymbol{a_T^*}^T)^T + \boldsymbol{g}S^*_{T,n}\boldsymbol{g}^T. \tag{10}$$

The error variance of a partial column-averaged mixing ratio state ($S^*$) can then be calculated from the covariance matrices

that represent the errors of the mixing ratio profiles ($\mathbf{S}$):

$$S^* = \mathbf{w}^{*T}\mathbf{S}\mathbf{w}^*, \tag{11}$$





i.e. for calculating the noise error variances of the combined product, we use as $\mathbf{S}$ the noise covariance $\mathbf{S_n}$ of Eq. (10).

Figure 7 depicts $50 \times 50\,\mathrm{km}$ averages for DJF and JJA of the noise error for the different data products (i.e. the square-root-values of the variances obtained according to Eq. 11). The noise errors follow a spatial pattern that is similar to the pattern as observed in the DOFS values. If there is a low DOFS value, i.e. the retrieval product cannot capture the real atmospheric $CH_4$ variations well, the noise error is relatively large. The noise error shows a similar latitudinal dependency for $XCH_4$ and uts$XCH_4$ as the DOFS values, and a similar surface elevation (surface pressure) dependency for uts$XCH_4$ and tro$XCH_4$ as the DOFS values.

Concerning $XCH_4$, the noise error is smaller than $5\,\mathrm{ppb}$, except for a few locations of winter hemispheric high/middle latitudes. The uts$XCH_4$ noise error is about $12\,\mathrm{ppb}$ at low latitudes and up to $20\,\mathrm{ppb}$ at high latitudes. For tro$XCH_4$, the noise error is about $10\,\mathrm{ppb}$ over the ocean and land with low surface elevation, but it can reach $20\,\mathrm{ppb}$ over land with high surface elevations (e.g. Himalaya, Andes).

## 5.2 Dislocation error

The dislocation error covariance matrix is calculated by

$$\mathbf{S_{dl}} = \mathbf{A_{dl}} \mathbf{S_{\Delta_{dl}}} \mathbf{A_{dl}}^T, \tag{12}$$

where $\mathbf{A_{dl}}$ is the dislocation kernel and $\mathbf{S_{\Delta_{dl}}}$ is the covariance matrix for the $CH_4$ dislocation uncertainty. More details on the dislocation kernel and the dislocation uncertainty covariance are given in Appendix E of Schneider et al. (2022b). From the dislocation error covariances, we calculate the dislocation variances for the layers representing the $XCH_4$, uts$XCH_4$, and tro$XCH_4$ data products according to Eq. (11). Figure 8(a-e) depicts the square-root-values of these variances in terms of averages for the DJF and JJA seasons. The dislocation error tends to be larger in the southern hemisphere than in the northern hemisphere, which reflects the latitudinal gradient of the temporal mismatch (see Fig. 2). Moreover, we observe locations with high surface elevation (Himalaya, Andes, Rocky Mountains, inner Greenland, Antarctica) have a more substantial tro$XCH_4$ dislocation errors compared to the surrounding region. This is caused by the small depth of the respective tropospheric partial layer. Then, the first kilometer above ground where the dislocation uncertainties are highest cover almost the full tropospheric partial column layer. Overeall, the dislocation errors are significantly smaller than the noise errors (compare Figs. 7 and 8).

## 6 The added value of tro$XCH_4$

In the previous sections, we documented the coverage and the characteristics of the data products. This section discusses the different patterns observable in the three data products: the $XCH_4$ and uts$XCH_4$ products, which have have a similar characteristics as the respective individual TROPOMI $XCH_4$ and IASI uts$XCH_4$ products, and the tro$XCH_4$ product, which is exclusively available in the combined data products (there is no respective TROPOMI or IASI data product). In the following, we illustrate the value that the tro$XCH_4$ product adds to the $XCH_4$ and uts$XCH_4$ products.





## 6.1 Global CH$_4$ patterns

Figure 9 (a-c) shows the JJA $50 \, \text{km} \times 50 \, \text{km}$ averages for XCH$_4$, utsXCH$_4$ and troXCH$_4$, and Fig. 9 (d-f) the respective DJF averages. There is a systematic difference between the typical XCH$_4$, utsXCH$_4$ and troXCH$_4$ values. Typical values are 1750-

1900 ppb for XCH$_4$, 1650-1875 ppb for utsXCH$_4$, and 1800-2000 ppb for troXCH$_4$. The decrease of the typical values, from troXCH$_4$ over XCH$_4$ to utsXCH$_4$, is caused by the decrease of the CH$_4$ concentrations above the tropopause, which is generally located at altitudes above the 300-400 hPa pressure level. The troXCH$_4$ product are the averaged mixing ratios for the layer from the ground to 450 hPa, i.e. a layer that is not affected by the decrease at high altitudes. The total column averages (XCH$_4$) are partly affected by this decrease, and the averaged mixing ratios for a layer representing high altitudes (utsXCH$_4$ represents

all pressure levels below 450 hPa) are strongly affected by this decrease.

The tropopause altitudes are lowest at high latitudes (around the 300-400 hPa pressure level) and highest in the tropics and subtropics (around the 100 hPa pressure level). This latitudinal gradient is mainly responsible for the gradients observed in the utsXCH$_4$ data (Fig. 9(b, e)). In the tropics and subtropics, where a large part of the UTS layer (Upper Troposphere/Stratosphere layer, 450 hPa - top of atmosphere) is situated below the tropopause, the utsXCH$_4$ values are rather high (1800-1900 hPa). Vice

versa, at middle and high latitudes, a significant part of the UTS layer is above the tropopause and consequently the utsXCH$_4$ values are low (1600-1750 ppb).

As aforementioned, the troXCH$_4$ product is not significantly affected by the tropopause altitude, and consequently the horizontal patterns, as seen in the troXCH$_4$ maps (Fig. 9(c, f)) are significantly different from the utsXCH$_4$ patterns. For troXCH$_4$, we see a clear gradient with increasing values from the Southern to the Northern hemisphere. This gradient is most

pronounced in the map showing the DJF averages (i.e. in the season when CH$_4$ concentrations peak in the Northern hemispheric troposphere (e.g. Frankenberg et al., 2005)). Moreover, there are some troXCH$_4$ hotspots, i.e. locations where the mixing ratios are larger than in its surrounding (e.g. tropical Africa and tropical South America, Northern India, and North-Eastern China), which are areas with high natural and/or anthropogenic CH$_4$ emissions (e.g. Saunois et al., 2016). In this context, the global troXCH$_4$ patterns are directly related to global CH$_4$ emission patterns.

Mixing ratios averaged over the total column (i.e. the XCH$_4$ product) should represents lower tropospheric and tropopause altitude related CH$_4$ signals. This is evident in Fig. 9(a, d), which demonstrates that the global pattern observed in the XCH$_4$ data product results from a superposition of the global tropopause altitude distribution and the global CH$_4$ emission pattern.

## 6.2 Local CH$_4$ time series

Figure 10 shows a time series for Madrid ($39.42°$N - $41.42°$N and $4.70°$W - $2.70°$W) of January 2018 to the June 2021 of

the combined data products. The first row shows the combined methane products (XCH$_4$, utsXCH$_4$, and troXCH$_4$, Fig. 10(a-c)). The utsXCH$_4$ time series shows the strongest seasonal cycle, with a maximum during the end of summer/autumn and a minimum in winter/spring. This observation is explained by the respective seasonal cycle of the tropopause altitude (see also the discussion in the context of Fig. 9). Seasonal cycle signal are much weaker in the XCH$_4$ and troXCH$_4$ time series. The XCH$_4$ product is still weakly affected by the seasonal cycle of the tropopause altitude. It shows a maximum during the end





of summer and autumn, but the minimum in winter/spring is hardly observable. This is due to the seasonal cycle of methane in the lower troposphere, being characterised by a maximum in winter/spring. This winter/spring maximum is actually the dominating seasonal cycle signal in the troXCH$_4$ time series (see Fig. 10(c)).

In Fig. 10(a-c), we mark data that might be affected by higher uncertainties, a snow covered surface (data marked by orange colour) or due to strong aerosol scattering (data marked by blue colour). Lorente et al. (2021) documents larger TROPOMI

XCH$_4$ uncertainties for observations over ground covered by snow. These observations can be identified by the so-called blended albedo ($A_b$) being larger than $0.85$ (Wunch et al., 2011):

$$A_b = 2.4 A_{\mathrm{NIR}} - 1.13 A_{\mathrm{SWIR}}. \tag{13}$$

Furthermore, scattering by aerosols and cirrus particles can cause high errors in the TROPOMI XCH4 product if not appropriately taken into account. Butz et al. (2012) introduced the parameter

$$C_s = \frac{\tau_s z_s}{\alpha_s} \tag{14}$$

and suggests to filter out observations with $C_s$ above $120\,\mathrm{m}$, because of significant aerosol scattering and thus potentially large uncertainties in the TROPOMI XCH$_4$ product ($\tau_s$, $z_s$, and $\alpha_s$ are estimated by the TROPOMI retrieval code and represent the optical aerosol thickness, the centre height of the aerosol layer, and the aerosol size parameter, respectively).

The second row of Fig. 10 (d-f) shows the a priori data simulated by the TM5 model. The a priori data show much less small

scale variability than the retrieved data and enable a clear identification of the different seasonal cycles in the troposphere and the upper troposphere/stratosphere.

The third row of Fig. 10 (g-i) shows the difference of the retrieved values and the a priori data ($\Delta\mathrm{CH}_4 = \mathrm{CH}_4 - \mathrm{Apriori}\mathrm{CH}_4$). The data that might be affected by snow covered surface or strong aerosol scattering have been filtered out. The large scatter indicates that on small scales the a priori model can differ significantly from the observations. Since these differences are

significantly larger than the uncertainties of the observations, the observations seem to provide a lot of information about small-scale processes that is not captured by the TM5 a priori model. In particular in the utsXCH$_4$ and troXCH$_4$ time series, we can also observe signals on a seasonal scale, that are beyond the uncertainties of the observations. This suggests that the observations contain also information about large-scale processes that are not captured by the model. In this context, the separation into utsXCH$_4$ and troXCH$_4$ gives valuable additional information. For instance, in the beginning of 2019 the $\Delta\mathrm{CH}_4$

values for the upper troposphere/stratosphere and the troposphere differ systematically from zero, but they are not correlated, i.e. this discrepancy between model and observations is less visible in the XCH$_4$ data and can be much better identified in the combined utsXCH$_4$ and troXCH$_4$ products.

### 6.3  Regional CH$_4$ patterns

Figure 11(a-c) depicts 3.5 year average (January 2018 – June 2021) of XCH$_4$, utsXCH$_4$, and troXCH$_4$ on a $0.1° \times 0.1°$ grid

with a zoom on the Iberian Peninsula and Fig. 11(d) shows a respective map of the EDGAR v7.0 anthropogenic emissions catalogue (representative for 1970-2022, Crippa et al., 2024). We only plot the XCH$_4$, utsXCH$_4$, and troXCH$_4$ averages when





at least 10 individual observations are availble for the respective $0.1° \times 0.1°$ grid box, which largely explains the locations with no data in Fig. 11(a-c). This area has also been used in the study of Tu et al. (2022) for studying landfill methane emission from ground and space.

Concerning the $XCH_4$ and $utsXCH_4$ maps, we observe a clear north-south gradient. In the northern part of the Peninsula, $XCH_4$ is generally below and in the southern part above $1860\,ppb$. Similar for $utsXCH_4$, which is mainly below $1780\,ppb$ in the north and above $1780\,ppb$ in the south. Exceptions are the coast lines, and the Ebro valley (south of the Pyrenees), for which $XCH_4$ values are also occasionally above $1860\,ppb$ in the north. The pronounced north-south gradient in the 3.5 year average is caused by the climatology of the tropopause altitude. Climatologically (average over 3.5 years) the tropopause is

significantly lower (at higher pressure levels) in the north than in the south.

The horizontal structures of the $troXCH_4$ map are significantly different from the structures in the respective $XCH_4$ and $utsXCH_4$ maps. We do not observe a significant north-south gradient. The $troXCH_4$ is below the tropopause, even for extreme cases when it might be situated at 300-400 hPa. Consequently, the $troXCH_4$ values are independent from the strong $CH_4$ signals introduced by the location of the tropopause. We observe highest $troXCH_4$ values (above $1940\,ppb$) in the Ebro valley,

in the center of the Peninsula in the area around Madrid, and along the coast lines. These $troXCH_4$ patterns are similar to the patterns present in the map of the anthropogenic EDGAR emissions inventories (Fig. 11d). This is a strong indication that the $troXCH_4$ product (only obtained by combining TROPOMI and IASI products) offers a much better possibility for monitoring the anthropogenic emissions than the $XCH_4$ and $utsXCH_4$ products, obtainable by TROPOMI and IASI alone.

## 7    Data quality recommendations

In this section, we give recommendation for filtering out data of reduced quality. We recommend filters for data that have an atypically large uncertainty caused by an atypically large noise of the original IASI and/or TROPOMI L2 data products or by a significant dislocation of IASI and TROPOMI (these errors are discussed in Sect. 5). Moreover, we recommend filters for data, where the TROPOMI observation might be strongly affected by surface snow cover and/or scattering by aerosols (these errors are discussed in Sect. 6.2).

For about 1% of the all combined data, we estimate a noise error for $XCH_4$ of more than $5\,ppbv$ and for $utsXCH_4$ and $troXCH_4$ of more than $20\,ppbv$. We recommend to use these values as thresholds for filtering out data of particularly poor quality. The dislocation error is significantly smaller, but can also affect the data quality. For about 1% of all the combined data products, we estimate a dislocation error for $XCH_4$ of more than $2\,ppbv$ and for $utsXCH_4$ and $troXCH_4$ of more than $15\,ppbv$. In order to avoid a significant impact of the dislocation error, we recommend to remove the respective data.

In Sect. 6.2, we introduce the blended albedo ($A_b$) and the aerosol parameter ($C_s$) for identifying TROPOMI data that might be significantly affected by surface snow cover and/or scattering by aerosols. In order to avoid an impact of these uncertainities on the combined data products, it is recommended to filter out data with $A_b \geq 0.85$ and with $C_s \geq 120\,m$. Figure 12 illustrates the percentage of data rejection by these snow cover and aerosol scattering filters. Panels (a) and (c) depict data rejection by the $A_b$ filtering, while panels (b) and (d) show data rejection by the aerosol scattering filter for the JJA and DJF seasons,





respectively. The data rejections due to strong aerosol scattering is most prominent in the JJA season in the subtropics in North Africa, Southern Europe and Asia. These are the (semi-) desert areas which suggest that mineral dust aerosols can importantly impact on the TROPOMI data quality. The data rejection due to snow covered surfaces is very important over the polar regions and over the northern hemispheric middle latitudes in winter (DJF season). The quality filter recommendations are resumed in Table 2.

**Table 2.** Table summarizing the recommendations for filtering out data of relatively poor quality. Data corresponding to here given value range should be filtered out for ensuring highest data quality.

| Filter | Value range for data of poor quality |
| --- | --- |
| Noise error | $\geq 5\,\text{ppbv}$ (for $XCH_4$); $\geq 20\,\text{ppbv}$ (for $utsXCH_4$ and $troXCH_4$) |
| Dislocation error | $\geq 2\,\text{ppbv}$ (for $XCH_4$); $\geq 15\,\text{ppbv}$ (for $utsXCH_4$ and $troXCH_4$) |
| Snow/ice cover | $A_b \geq 0.85$ (see Eq. 13) |
| Aerosol scattering | $C_s \geq 120\,\text{m}$ (see Eq. 14) |

## 8   Summary


We apply the Schneider et al. (2022b) synergetic data combination method to the large TROPOMI and IASI $CH_4$ data sets. We generate a combined data consisting of about 289 million data points, that are globally distributed and representative for the 42 month between January 2018 and June 2021. The combined data set consists of three different data products: total column-averaged mixing ratios ($XCH_4$), upper tropospheric/stratospheric column-averaged mixing ratios ($utsXCH_4$), and tropospheric

column-averaged mixing ratios ($troXCH_4$). Whereas the former two have a very similar characteristics and quality as the respective TROPOMI and IASI data products, the latter is a unique outcome of the synergetic combination. High-quality and reliable $troXCH_4$ data can only be achieved by combining the two data sets optimally. This data product is neither available from TROPOMI nor from IASI alone.

We show that this $troXCH_4$ data product has a scientific impact on top of the $XCH_4$ and $utsXCH_4$ data products. While $XCH_4$

and $utsXCH_4$ are significantly affected by the strong $CH_4$ variations in upper troposphere and stratosphere (for instance, due to variations in the tropopause altitude), the $troXCH_4$ data product is strongly connected to the surface $CH_4$ emissions. Thus it offers important advantages over the individual TROPOMI and the IASI data products when it comes to the research and monitoring of anthropogenic $CH_4$ emissions.

The merged data set is publicly-accessible and it can be can be downloaded in the form of a NetCDF file per day of ob-

servations from our servers at KIT: https://www.imk-asf.kit.edu/english/CH4-synergy-IASI-TROPOMI_RemoTeC.php. The respective NetCDF data files comply with the FAIR principles (Wilkinson et al., 2016), i.e. among others, each data point is provided with its respective error and averaging kernel.



The method can easily be applied to other TROPOMI and IASI data sets, as long as these data are made available un-
der consideration of the FAIR principles (the availability of individual errors and averaging kernels is mandatory). Moreover,
the computational efficiency and flexibility of the method (automatic benefit from novel developments made by dedicated
TROPOMI and IASI retrieval experts) will become important, particularly in the context of the Metop-SG (Second Genera-
tion) satellite mission (https://www.eumetsat.int/metop-sg), which is planned to start by the end of 2025. Metop-SG will have
TROPOMI and IASI successor instruments aboard and offer an unprecedented high number of diverse and high-quality $CH_4$
data that can be used for a synergetic data combination.

*Data availability.* For review, the data are accessible at https://radar.kit.edu/radar/en/dataset/wq583rnzpmd83m5g?token=UEuECSWHlGgWBdoPVsvI
(Shahzadi et al., 2025) and made freely available at https://www.imk-asf.kit.edu/english/CH4-synergy-IASI-TROPOMI_RemoTeC.php.

*Author contributions.* KS and MS performed the data merging and prepared the manuscript. UC and JM supported the development of the
geomatching algorithm. KS, MS, NYL and FH were involved in generating the MUSICA IASI data set. TB and MMV were involved in
generating the RemoTeC TROPOMI data set. All authors supported the generation of the final version of the manuscript.

*Competing interests.* The authors have no competing interests.

*Acknowledgements.* This research has been supported by the National High-Performance Computing (NHR) alliance via its member Scien-
tific Computing Center (SCC) at Karlsruhe Institute of Technology (KIT). The respective NHR@KIT project IDs are UFOS and LASSIE.
We acknowledge the support of the Initiative and Networking Fund of the Helmholtz Association via the Helmholtz Metadata Collaboration
project "Metamorphoses" (funding ID: ZT-I-PF-3-043). Important part of this work was performed on the supercomputer HoreKa funded
by the Ministry of Science, Research and the Arts Baden-Württemberg and by the German Federal Ministry of Education and Research.
We acknowledge the support by the Deutsche Forschungsgemeinschaft and the Open Access Publishing Fund of the Karlsruhe Institute of
Technology.

## Figures

**Input data products**

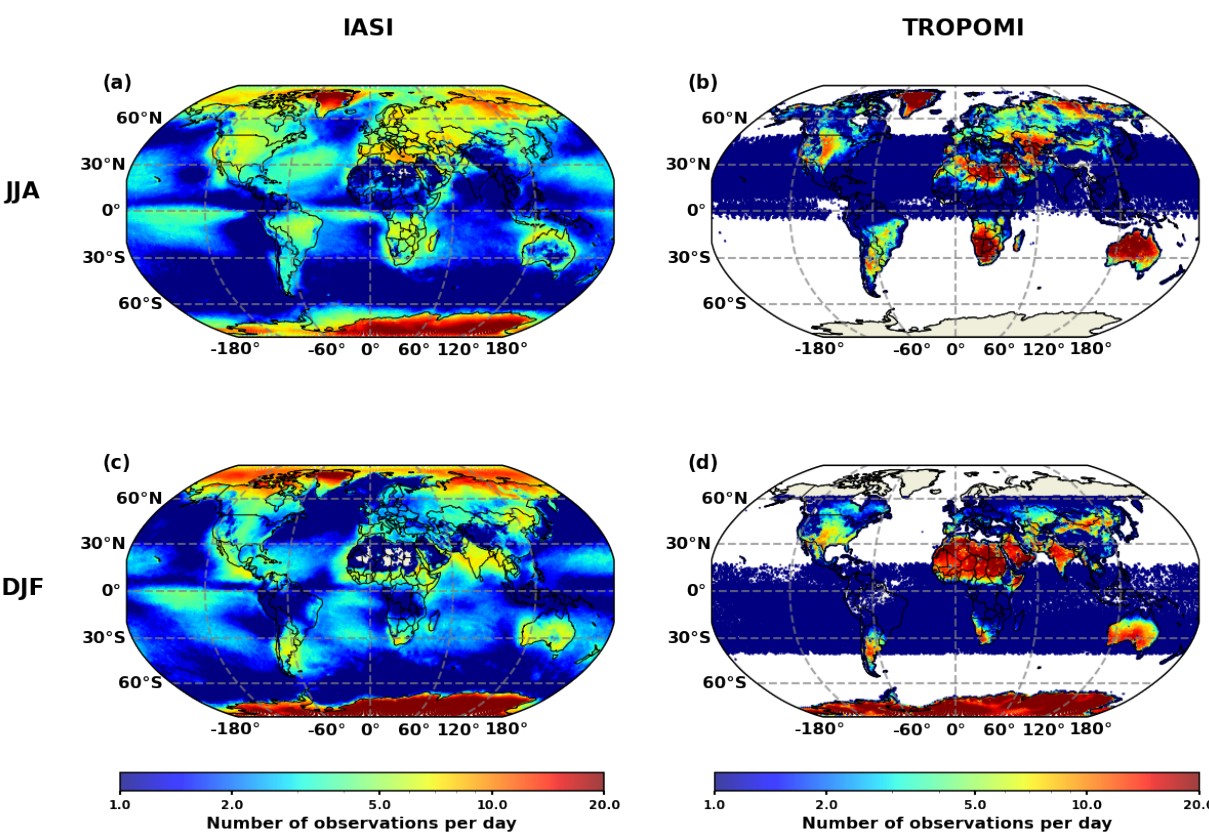

**Figure 1.** Average daily number of observations per $50\,\text{km} \times 50\,\text{km}$ grid box for June, July, and August (a, b) and for December, January, and February (c, d) for IASI and TROPOMI, respectively.





**Geomatching**

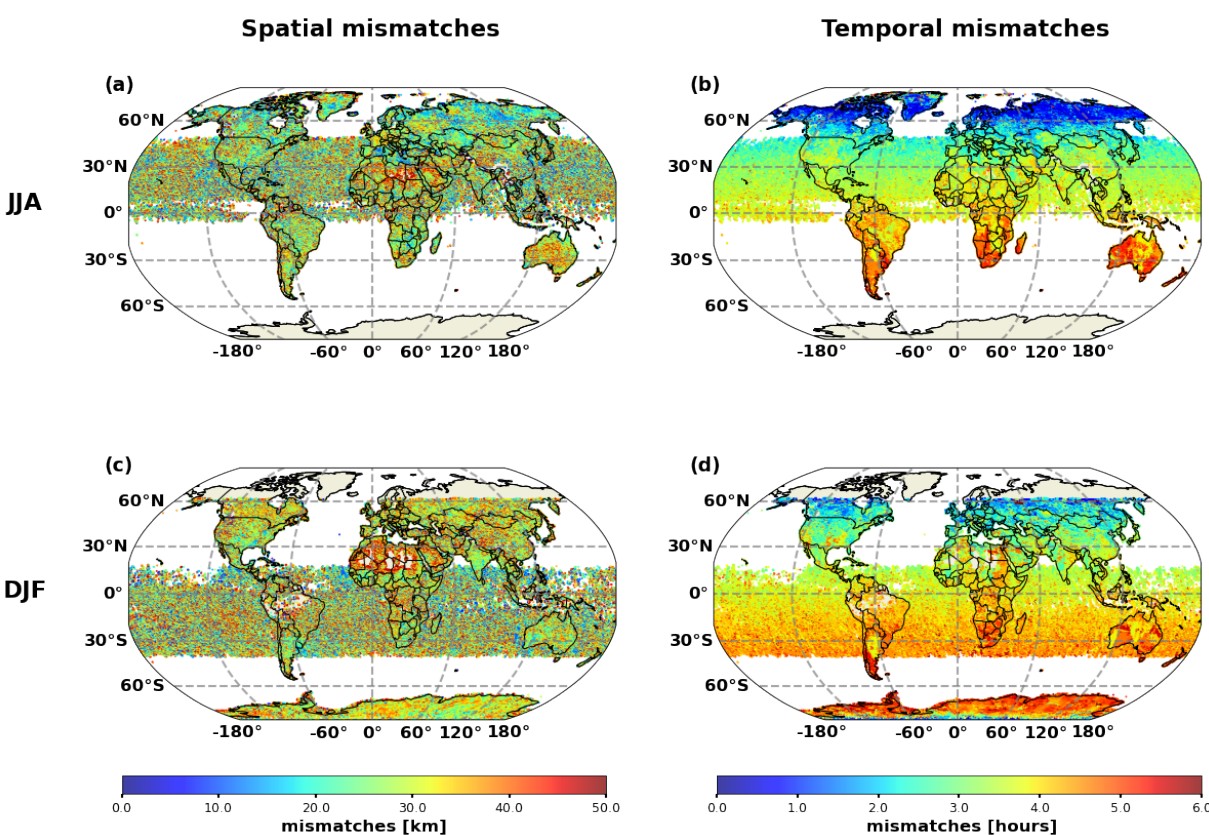

**Figure 2.** Mean spatial and temporal mismatches per 50 km×50 km grid box for June, July and August in (a, b) and for December January and February in (c, d) respectively of the combined product.

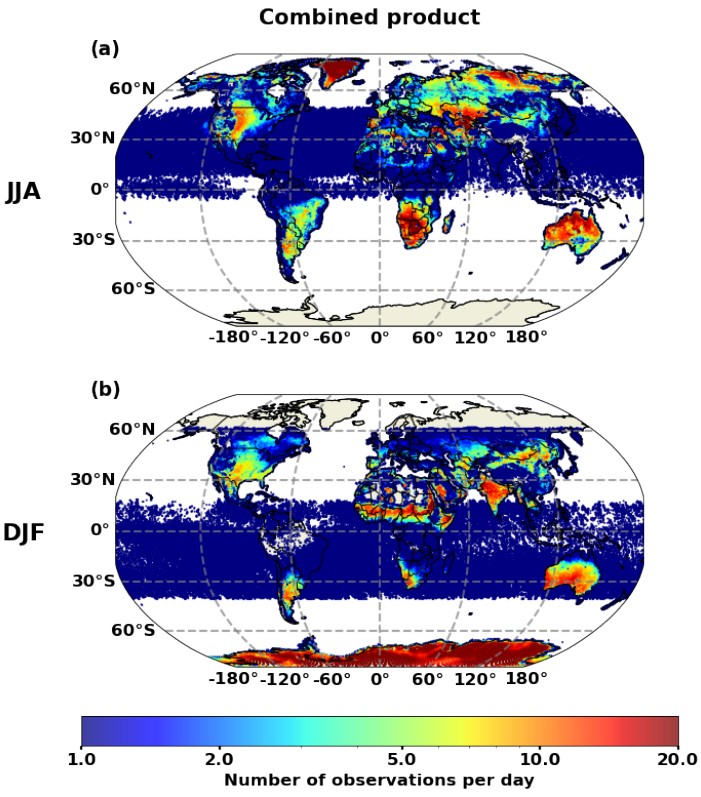

**Figure 3.** Average daily number of observations per $50\,\mathrm{km} \times 50\,\mathrm{km}$ grid box for: June, July, and August in (a), and December, January, and February in (b) of the combined product.

## Averaging kernels

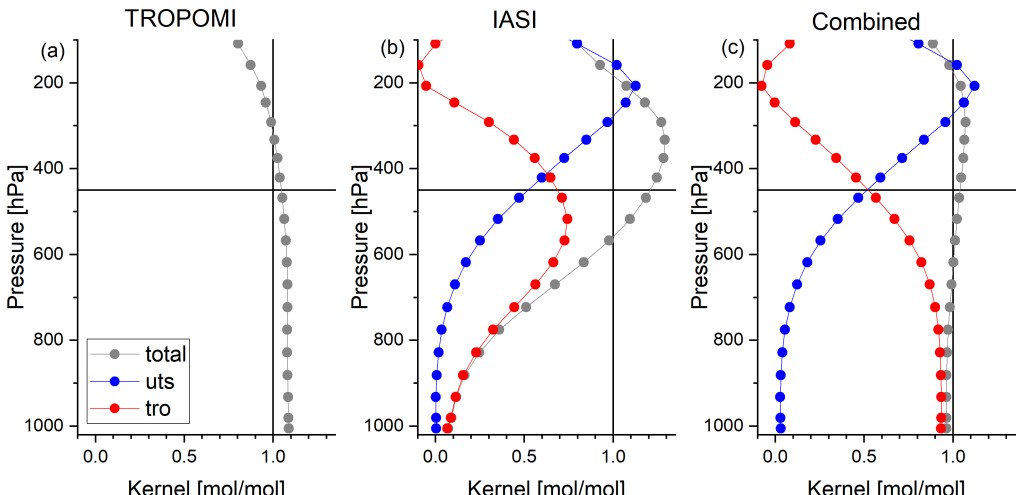

**Figure 4.** Total and partial column amount kernels for (a) TROPOMI, (b) IASI, and (c) Combined product. Grey: total column amount kernel; Blue: upper tropospheric and stratospheric partial column amount kernel (utsXCH$_4$, 450hPa - T.A.O); Red: tropospheric partial column amount kernel (troXCH$_4$, surface – 450hPa).





## Global sensitivity patterns

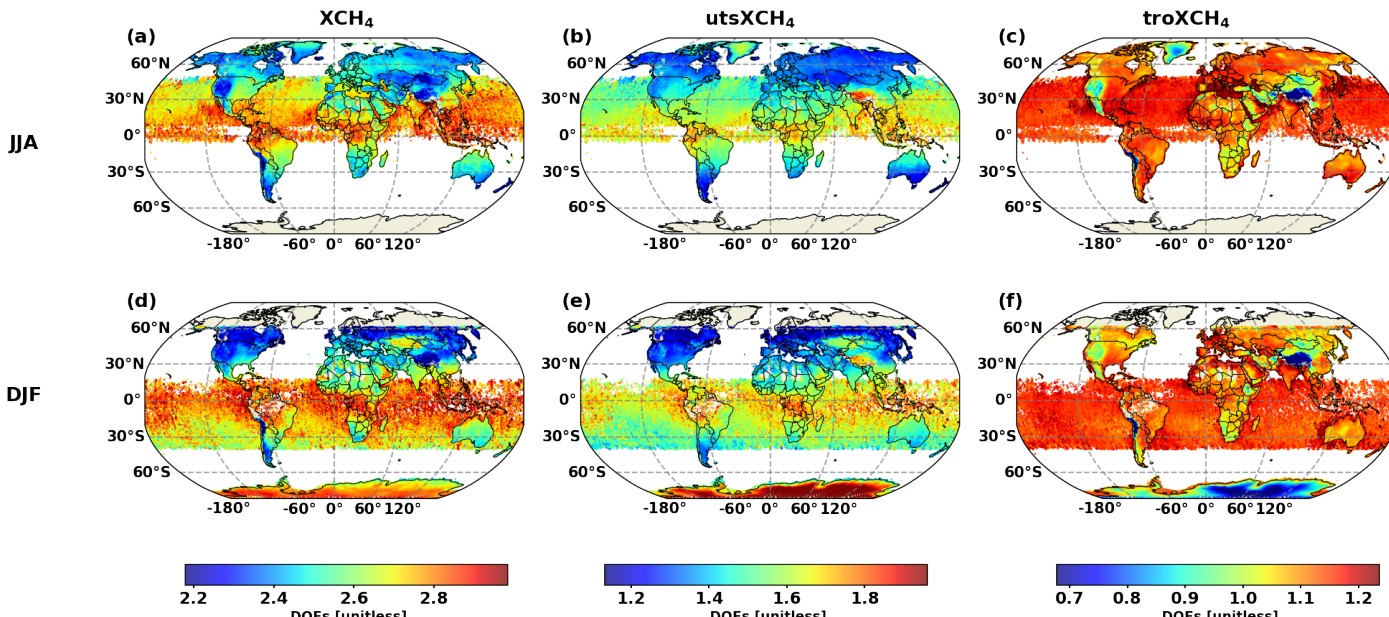

**Figure 5.** Degree of Freedom of Signal (DOFS) of the combined products averaged for a $50\,\text{km} \times 50\,\text{km}$ grid box. For the total column (XCH$_4$, a+d), the upper troposphere and lower stratosphere (utsXCH$_4$, b+e), and the troposphere (troXCH$_4$, c+f). For June, July and August (JJA, a-c), and December, January and February (DJF, c-f).



## Vertical representativeness

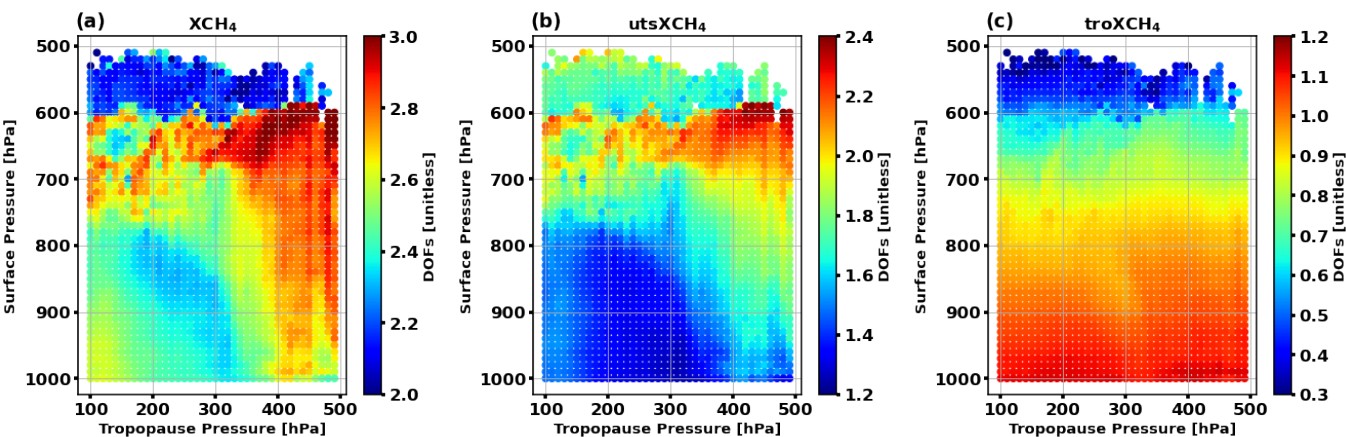

**Figure 6.** Degree of Freedom of Signal (DOFS) distribution as a function of surface pressure and tropopause pressure: Total column DOFS (a), utsXCH$_4$ DOFS (b), and troXCH$_4$ DOFS (c).

**Noise error**

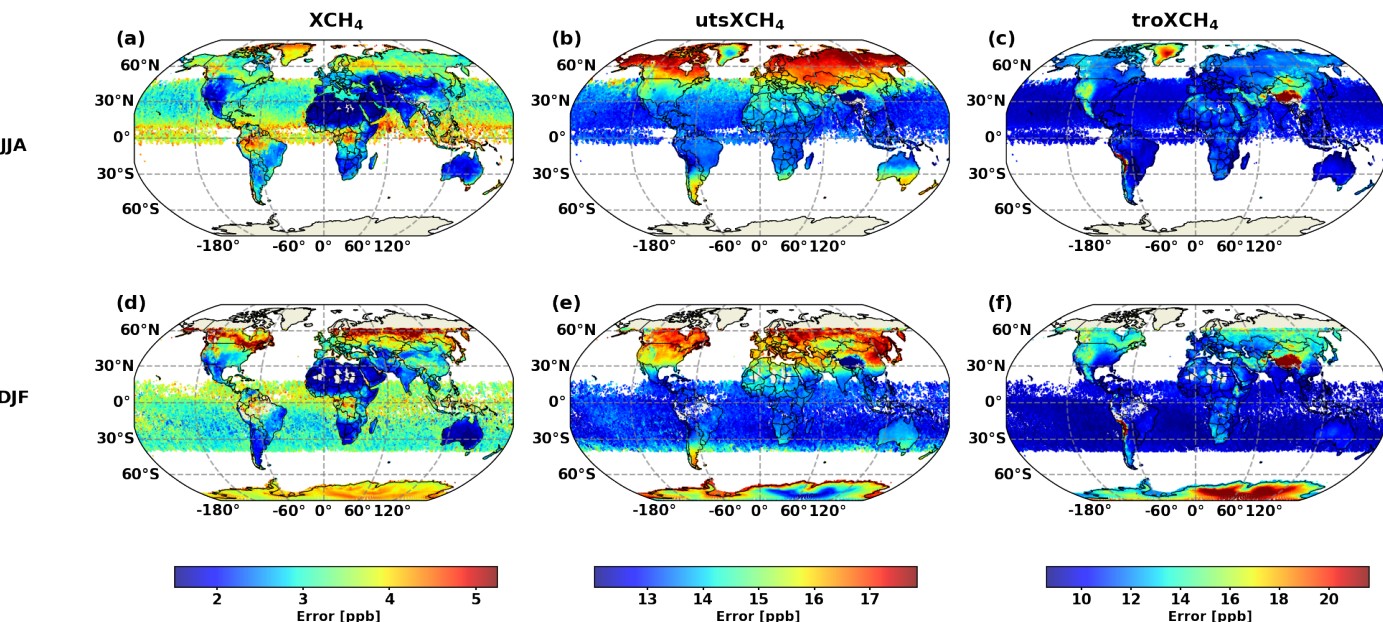

**Figure 7.** Noise error of the combined products averaged for a $50\,\mathrm{km}\times50\,\mathrm{km}$ grid box. For the total column in (XCH$_4$, a+d), for the upper troposphere and stratosphere (utsXCH$_4$, b+e) and for the lower troposphere (troXCH$_4$, c+f). For June, July and August (JJA, a-c), and December, January and February (DJF, c-f).





**Dislocation error**

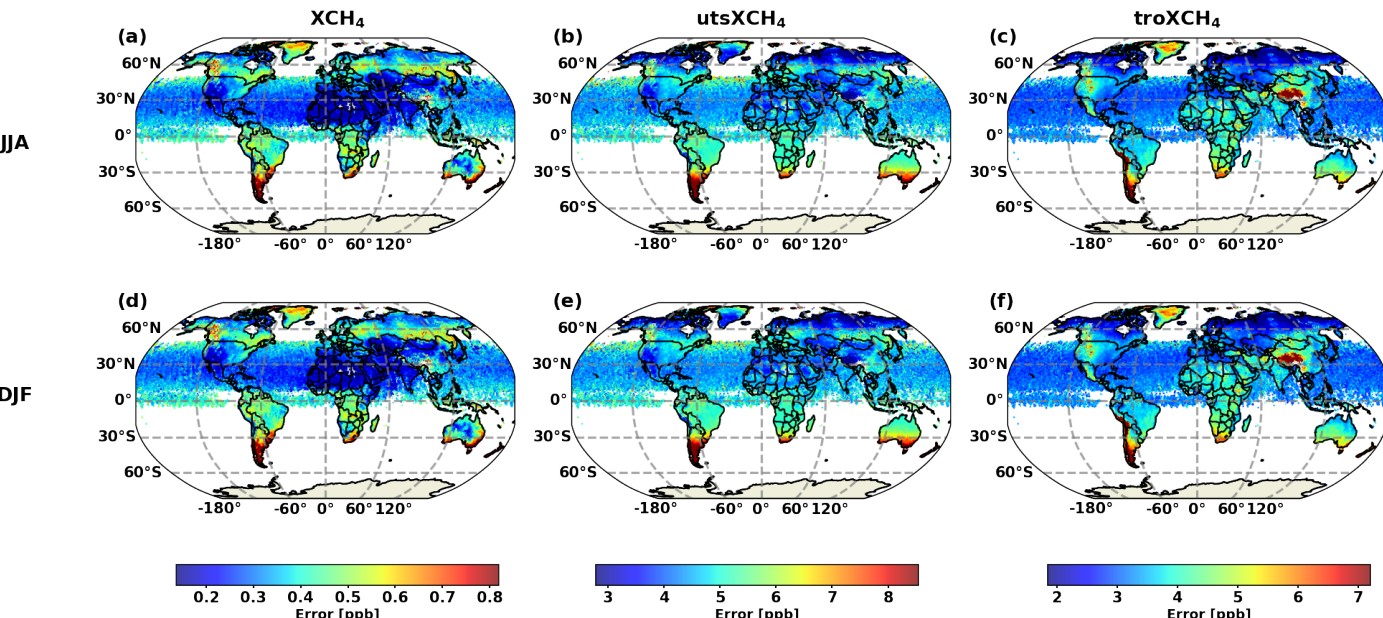

**Figure 8.** Same as Fig. 7, but for the dislocation error.



**Global CH4 patterns**

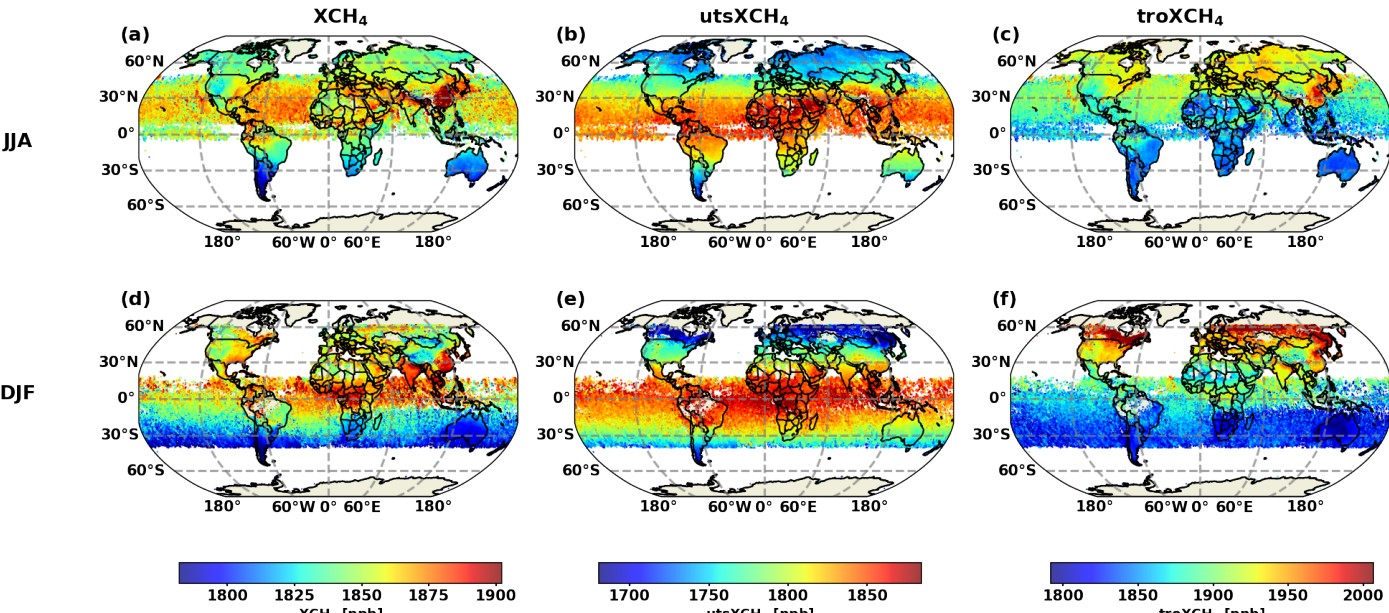

**Figure 9.** Global average methane data of the combined products averaged for $50\,\text{km} \times 50\,\text{km}$ grid box. For the total column (XCH$_4$, a+d), for the upper troposphere and stratosphere (utsXCH$_4$, b+e) and for the lower troposphere (troXCH$_4$, c+f). For June, July and August (JJA, a-c), and December, January and February (DJF, c-f).



## Local CH4 time series

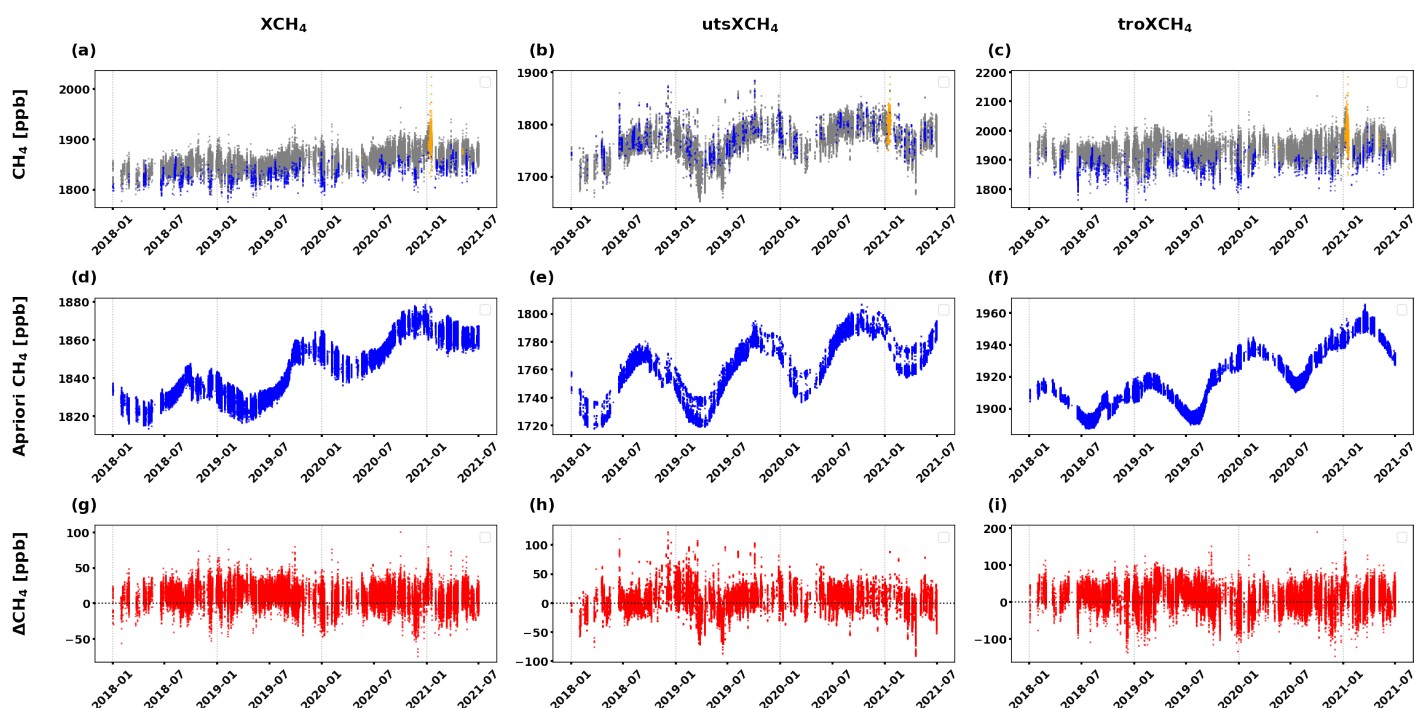

**Figure 10.** Time series data (January 2018 - June 2021) for a $2° \times 2°$ area around Madrid. (a-c): Retrieved products (orange color indicates data points where the blended albedo (A_b) $\geq 0.85$, blue highlights where the TROPOMI aerosol parameter $\geq 120$, and grey represents all other data points); (d-f) The a priori data. (g-i): the difference between the retrieved and the a priori data. (a),(d), and (g) for $XCH_4$; (b), (e), and (h) for $utsXCH_4$; (c), (f), and (i) for $troXCH_4$.



**Earth System Science Data Discussions**

**Regional CH4 patterns**

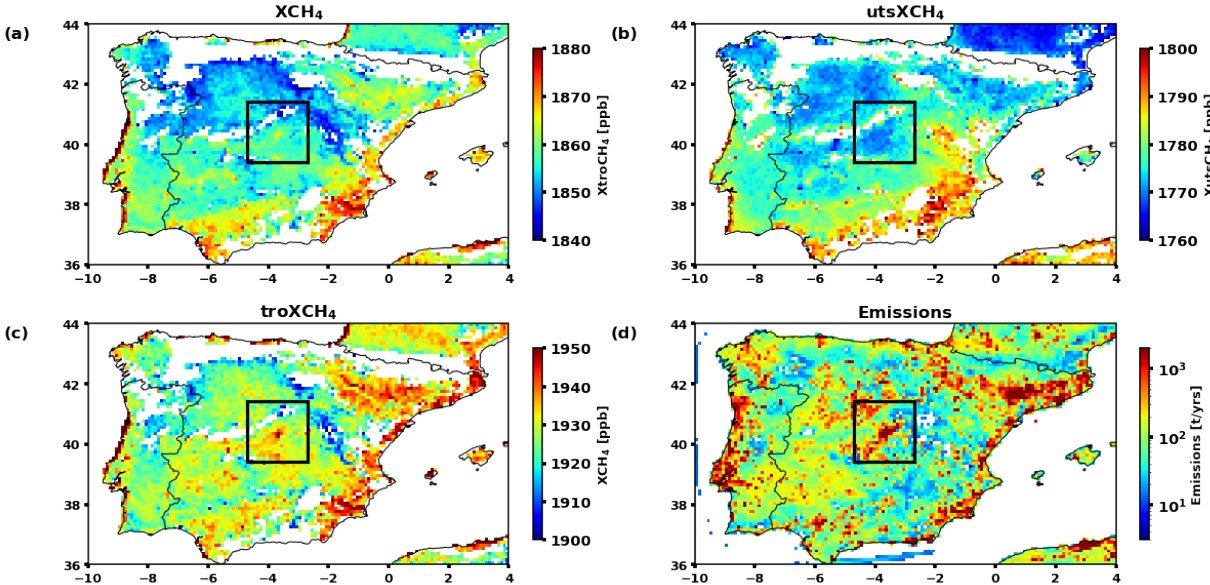

**Figure 11.** Averaged methane data for the Iberian Peninsula. For the total column (XCH$_4$, a), for the upper troposphere and stratosphere (utsXCH$_4$, b), and for the lower troposphere (troXCH$_4$, c). Panel (d) shows the anthropogenic emissions of EDGAR v8.0 GHG data (1970–2022). The black box indicates the $2° × 2°$ area around Madrid used in the context of Fig. 10.


**Data filters**

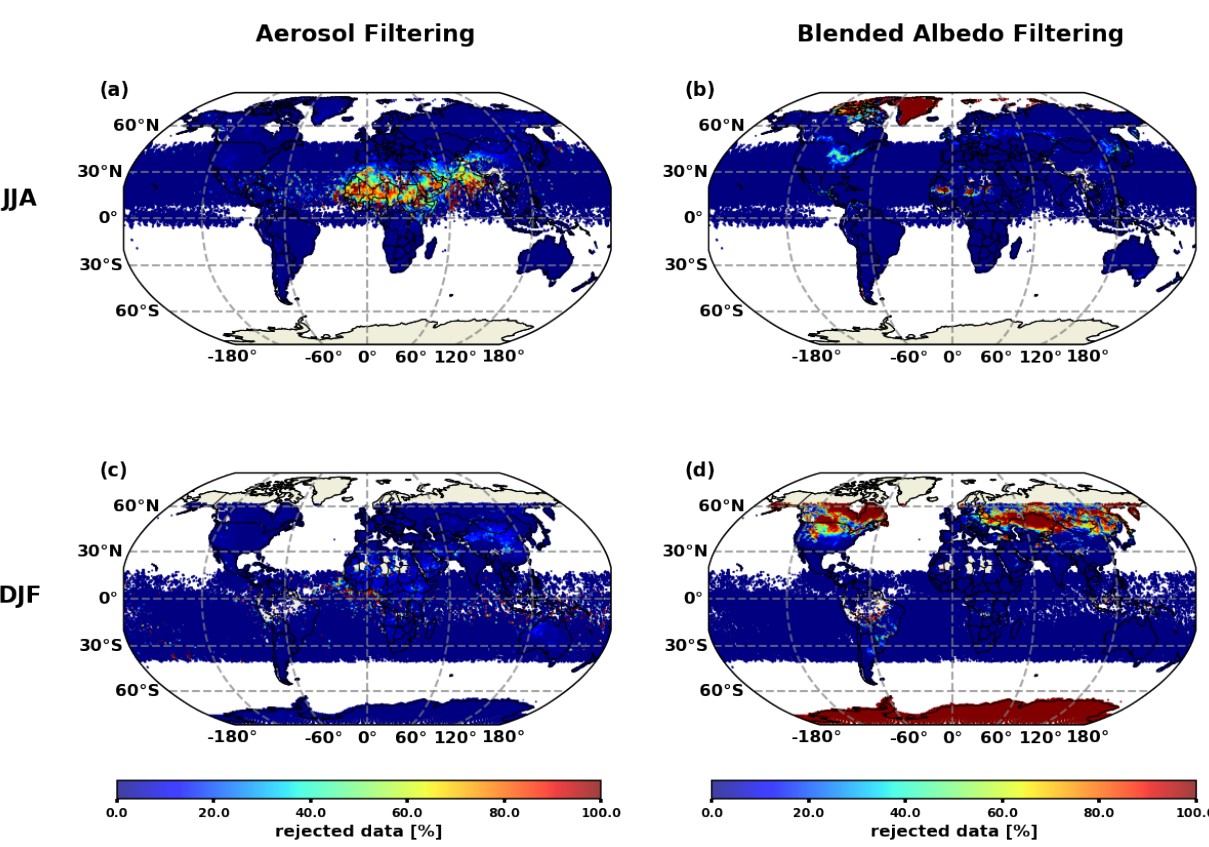

**Figure 12.** Relative number of data rejected in a $50\,\mathrm{km} \times 50\,\mathrm{km}$ grid box by the aerosol scattering filter in (a,c) and by the snow coverage filter (b,d). For June, July and August (JJA, a, b) and for December, January and February (DJF, c, d) respectively.





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
