# Peer review of "A multi-year global methane data set obtained by merging observations from TROPOMI and IASI"

_Earth System Science Data, 2025_

## Author Comment (AC1)

This is a review of the manuscript titled "A multi-year global methane data set obtained by merging observations from TROPOMI and IASI" submitted to ESSD by Shahzadi et al. The authors use the algorithms proposed by Schneider et al. to produce a daily, global, long-term methane dataset from TROPOMI and IASI satellite retrievals. Overall, this work continues their previous studies, fits the scope of the journal, and is timely because methane has recently risen sharply and exerts a strong warming influence on the climate. Therefore, I recommend publication in ESSD after the authors address several issues related to key technical validation details and the presentation of results.

We thank the referee for their interest in our work and the effort spent in reviewing the manuscript. In the following, we reply to each of their comments and write our text in red fonts.

Main comments:

1. Although the authors have assessed the dataset accuracy using TM5 outputs and EDGAR emissions, ground measurements were not used for a more independent validation. I suggest a qualified comparison of the dataset with in situ/ground based results (e.g., TCCON).

We agree that the validation of new data products is very important. However, please note that the validation of the method and the data product has been shown in great detail in Schneider et al. (2022b). Therein, we compare the data to independent references of TCCON, Aircore, and GAW. We will try to make it even clearer in the text that Schneider et al. (2022b) show a detailed validation study of the data product.

Moreover, data inter-comparisons are beyond the scope of a regular ESSD article. The aims and scopes of a regular ESSD article are (according to https://www.earth-system-science-data.net/about/aims_and_scope.html): "Articles in the data section may pertain to the planning, instrumentation, and execution of experiments or collection of data. Any interpretation of data is outside the scope of regular articles. Articles on methods describe nontrivial statistical and other methods employed (e.g., to filter, normalize, or convert raw data to primary published data) as well as nontrivial instrumentation or operational methods. Any comparison to other methods is beyond the scope of regular articles."

2. The manuscript states that IASI is "adjusted to TROPOMI a priori (TM5)", but it is not explicit whether this is (a) a re retrieval with a different a priori, (b) an additive/multiplicative post processing correction, or (c) a linearization/offset mapping. The precise operation matters because it affects bias propagation and degrees of freedom.

We do not perform additional retrievals, which would be computationally extremely expensive. The strength of the method is that it works with already retrieved L2 data products, which makes it computationally cheap, i.e., ideal for very large data sets. Moreover, by working with the L2 products, the method automatically benefits from the retrieval developments made by the dedicated TROPOMI and IASI retrieval experts.

We adjust the IASI a priori to the TROPOMI a priori according to Eq. (B13) of Schneider et al. (2022b), which is also well explained in Rodgers and Connor (2003, https://doi.org/10.1029/2002JD002299, therein Eq. 10), i.e., it is an additive post-processing correction:

$$\hat{x}' = \hat{x} + (A - I)(x_a - x_a').$$

Here $\hat{x}$ and $\hat{x}'$ are the original and the adjusted/modified retrieval states, accordingly, $A$ and $I$ are the averaging kernel and the identity matrix, respectively, and $x_a$ and $x_a'$ are the original and the modified a priori states, respectively. We will clarify this in the text.

3.   The Kalman update uses instrument posterior covariances as input. It is unclear how those covariances were pretreated (regularized, truncated, or inflated), whether they include forward-model uncertainty, and how numerical inversion (or pseudo-inversion) is handled for large matrices.

For combining the IASI and TROPOMI L2 data, we use the IASI product as the background (and use the IASI retrieval a posteriori covariance as the background covariance) and then update this background with the additional information provided by the TROPOMI observation. The IASI a posteriori background is calculated according to Eq. (A8) of Schneider et al. (2022b):

$$S_{\hat{x}} = (I - A)S_a,$$

i.e., it depends on the a priori covariences $(S_a)$ and the averaging kernels $(A)$. We calculate $S_a = R^{-1}$ from the constraint matrix used for the IASI retrieval (see Sect. 4.6 of Schneider et al., 2022a):

$$R = (\alpha_0 L_0)^T(\alpha_0 L_0) + (\alpha_1 L_1)^T(\alpha_1 L_1) + (\alpha_2 L_2)^T(\alpha_2 L_2),$$

where $L_0$, $L_1$, and $L_2$ are the Tikhonov constraint matrices (Tikhonov 1963, https://zbmath.org/?q=an:0141.11001) and $\alpha_0$, $\alpha_1$, and $\alpha_2$ are diagonal matrices that allow for a vertically dependent strength of the Tikhonov constraints. The inversion $S_a = R^{-1}$ is calculated via the Woodbury identity (Woodbury 1950, https://mathscinet.ams.org/mathscinet/article?mr=38136), i.e., there is an algebraic solution for this inversion.

Please note that our data combination method needs no other matrix inversions: we update the IASI profile product by adding a total column observation (one value only), and for calculating the respective Kalman gain, we only have to invert a scalar (see Eq. (4) of the manuscript). We will make clear in the text that the method needs no numerical matrix inversion (or pseudo-inversion).

4.   Displacement errors are crucial for tropospheric products and are only described by reference to prior work. The current manuscript lacks the exact formulas, parameter choices (time/space scales), and tests showing sensitivity to those choices.

We will add the equation for the dislocation kernel (Eq. E3 in Schneider et al., 2022b), then the set of equations will be complete.

Please note that we carefully estimated the dislocation error using CAMS data (Copernicus Atmospheric Monitoring Service data). All this work is documented in detail in Appendix E of Schneider et al. (2022b), including figures showing how the atmospheric CH4 fields vary in

space and time. This detailed explanation of the dislocation error is just one click away, and we think it is sufficient to give a clear reference to this Appendix. In order to avoid any doubt on how the dislocation errors are estimated/calculated, we will revise the manuscript and try to further improve the referencing to Schneider et al. (2022b).

5. The manuscript notes lower DOFS and increased noise in high-altitude areas, but does not quantify the implications for users (e.g., systematic smoothing, negative bias, or overconfidence).

A large noise error means that the data product can be subject to significant uncertainty, and a low DOFS value means that the data product sticks to the a priori information, i.e., data with a low DOFS value provides limited information on top of the a priori data.

For a low surface pressure (high surface elevation), the layer below 450 hPa is rather shallow, and the satellite data contains too little information to be sensitive to it (low DOFS) and to detect it with good precision (high noise error). In order to avoid the shallow layers for high surface elevation, we will revise the data set. For the revision, we will separate the atmosphere 50%/50%, i.e., the lower 50% of the atmosphere will be the tropospheric layer, and the upper 50% of the atmosphere will be the upper tropospheric/stratospheric layer. This will remove the "artificial" dependency of the troXCH$_4$ DOFS and errors on the surface elevation.

The data set contains detailed information about their uncertainty (noise, dislocation) and their characteristics (averaging kernel, DOFS) for each individual observation. We recommend using the averaging kernel as the observation operators whenever the data are used in flux inversion studies, and also consider uncertainties of the observations. This is the best way to ensure that the information provided by the observation can be correctly interpreted by the inversion system. We will briefly mention this in Section 7 of the manuscript and call this section "Data quality and data reusage recommendation".

Moreover, please note that Table 2 summarizes our filter recommendations for high data quality.

Minor / Technical issues

1. Add a detailed NetCDF variable list (names, units, dimensions), an explanation of averaging kernel format/shape, and a short example (Python/xarray snippet) showing how to read XCH4, troXCH4, the averaging kernel, uncertainty, and the quality bitmask. Provide a recommended QC selection for typical use.

We carefully write all the variables in a NetCDF file with standardized metadata (1.7 CF conform). All information about the variables (names, units, dimensions, etc.) is available in these metadata and can be easily accessed by standard NetCDF viewing tools (e.g., Panoply). The user can download the data and easily get information about the data structure by looking at the metadata. We think that an additional table with this metadata information is not needed.

2. The selection of matching windows (50 km / 6 h, then normalized distance minimization)

should be justified quantitatively. If possible, add sensitivity tests for different spatial/time windows (e.g., 25 km / 3 h, 75 km / 8 h) to show tradeoffs between coverage and displacement error.

The dislocation between TROPOMI and IASI in space and time is described in large detail in Appendix E1 of Schneider et al. (2022b). Figure E1 reveals that allowing for a dislocation in time of only 25 km or up to 100 km instead of 50 km will approximately halve/double the spatial displacement error. Concerning the dislocation in time, a 6-hour displacement causes an error that can already be compared to the error of a 100 km spatial displacement (compare Figs. E1a and b of Schneider et al., 2022b). However, the 6-hour displacement is needed to have sufficient matches in the southern hemisphere. The used matching windows of 50 km and 6 hours are a reasonable compromise: they ensure (1) that the dislocation errors are clearly smaller than the noise error, and (2) that we have sufficient matches at any location.

3. In the Methods section, define each symbol and its dimension immediately below the equation. For Eq. (4), (5), (6), and (9), clarify matrix sizes and transpose conventions, and explicitly state when operations are performed in log space versus linear space (you mention omitting log-transform details for brevity, please explicitly state in Methods whether final arrays are reported on the linear or log scale).

We will work on clearly defining the symbols in order to avoid any doubts. Please note that all reported arrays (variables shown in the Figures, as well as the variables provided in the NetCDF file) are on a linear scale. The logarithmic scale is only used for the IASI data processing. We will make this clearer in the manuscript.

4. For Fig. 10, ensure aerosol/snow masking and other flags are clearly labeled in the panels or legend, not only in the title; please check and apply the same clarity to other figures as needed.

Yes, we will go through all figures and check for clarity.

---

## Author Comment (AC2)

This paper presents a global multi-year methane data product obtained by synergetically merging TROPOMI (XCH₄) and IASI (profile) Level-2 observations via a geo-matching step and a computationally efficient Kalman filtering scheme. The merged product provides three diagnostics—XCH₄, utsXCH₄ (≈450 hPa to TOA), and troXCH₄ (surface to 450 hPa)—with the stated goal of enhancing lower-tropospheric sensitivity and mitigating contamination from strong CH₄ gradients near the tropopause. The dataset covers Jan 2018–Jun 2021 with ~289 M combined samples, and the authors discuss uncertainty components (noise, dislocation/mismatch), DOFS patterns, and practical quality filters.

The contribution is valuable and timely. However, the physical and methodological explanation of why and how the Kalman synergy yields a lower-tropospheric partial column that is robust to tropopause height variability is under-developed in the current draft. The paper needs (i) clearer exposition of the state vector, observation operators, averaging kernels, and Kalman gain in the synergy; (ii) explicit sensitivity demonstrations showing reduced influence of UT/LS variability on troXCH₄; and (iii) several additional validations/robustness checks.

We would like to thank the referee for their interest and time spent in reviewing our manuscript. In the following, we write our replies to the comments of the referee as text in red fonts.

In comments #1, #5, #6, and #7, the referee points to the reduced troXCH₄ data quality (increased noise, reduced sensitivity) for elevated surface levels. This dependency of the troXCH₄ data quality is because of our choice of a fixed pressure level (450 hPa) for separating the surface-near atmosphere from the upper atmosphere. Naturally, for a high surface elevation, the surface-near layer up to this fixed pressure level becomes relatively shallow. It often becomes too shallow to be detectable at high quality using the signal provided by the satellite observations.

For the revised data set, we propose a separation into the lower 50% and the upper 50% of the atmosphere. This kind of separation avoids the strong degradation of the data quality for very shallow layers. Instead of trying to detect very shallow structures --- structures that are actually not detectable with the information available from the satellite measurements --- the 50%/50% separation used for the revised data set will focus on the quantities that are detectable. Please note that the fusion of IASI and TROPOMI data generates a full vertical profile. The separation into a surface-near and an upper atmospheric data product happens after the fusion calculations and should be chosen in a way that is most useful for the data users. Considering the comments of the referee, we think it is most useful to provide the data for layers that can be reasonably detected by the satellite measurements, i.e., a 50%/50% separation.

Moreover, we will provide the averaging kernels of the revised data set in the same vertical gridding as the original TROPOMI data product. We think that this facilitates data usage since the original TROPOMI data are already widely used in the community.

1. The manuscript claims and qualitatively illustrates that troXCH₄ (surface–450 hPa) is below the tropopause and therefore "independent from the strong CH₄ signals introduced by the location of the tropopause." This is a central selling point and should be

demonstrated more rigorously : Include row-wise AKs (and cumulative contribution functions) for several regimes (low vs. high tropopause, ocean vs. high terrain, clean vs. dusty/snow scenes). Demonstrate that *A* for the tro layer largely suppresses UT/LS influence, while the uts layer captures most tropopause-related variability. This will convert a qualitative assertion into a physical demonstration.

The partial column kernels shown in Fig. 4 are for mid latitudes. We will expand the figure and show example partial column averaging kernels in addition for low and high latitudes.

Moreover, in an additional Appendix, we will show the respective full vmr profile kernels and the respective Kalman gain profiles similar to Figs. 1 and 2 of Schneider et al. (2022b).

2. Justify the 450 hPa boundary physically (global climatology of tropopause heights, retrieval DOFS distribution, and typical IASI vertical sensitivity). Provide a sensitivity check with an alternative boundary (e.g., 500 hPa) to show that conclusions are not fine-tuned to a single threshold. (You note that even extreme tropopauses at 300–400 hPa leave troXCH$_4$ below the tropopause; quantify this across latitudes and seasons.)

The 450 hPa level has been chosen because been it is below the climatological tropopause for all seasons and locations. The climatological temperature tropopause height is between 17 km (about 150 hPa, tropics) and 7 km (about 420 hPa, Antartic summer), and the chemical tropopause is typically between 18km (about 100 hPa) and 10km (265 hPa). Both is derived from the CESM1–WACCM (Community Earth System Model version 1 – Whole Atmosphere Community Climate Model, Marsh et al., 2013, https://doi.org/10.1175/JCLI-D-12-00558.1) monthly output of 1979–2014 and documented in Fig. 5 of Schneider et al. (2022a).

However, the amount of information provided by the radiances measured by the satellite sensors depends on the quality of the satellite data (spectral noise, spectral resolution), and on the correct understanding of the spectroscopy (line intensities, line broadening, water continuum, etc.). The spectroscopic understanding and the measurement quality is good enough to separate the troposphere above 450 hPa from the atmosphere at lower pressures as long as the surface elevation is not too high. However, for a high surface elevation, the layer limited by the 450 hPa level becomes rather shallow and is difficult to retrieve from the available radiances. It cannot be detected clearly (lower DOFS) and only with large uncertainties (larger noise errors).

For the revised data set, we propose to separate surface-near and upper atmosphere at 50% of the surface pressure. This is practically the same as a separation at 450 hPa (for lowland, i.e., surface pressure above 800 hPa), but avoids very shallow and undetectable layers for a high surface elevation.

3. Add a controlled sensitivity test: perturb the a priori UT/LS by ±(50–100) ppb and show the response in XCH$_4$, utsXCH$_4$, and troXCH$_4$. I expect utsXCH$_4$ and XCH$_4$ to respond strongly, while troXCH$_4$ remains comparatively stable if the mechanism holds. Your Madrid/Iberia analysis already hints at this separation (XCH$_4$ shows superposition; troXCH$_4$ tracks near-surface seasonality/emissions), but a targeted experiment would make the case bullet-proof.

Yes, we agree. This additional test on global scale can give further insight and better document the usefulness of the data product.

We will document how shifts of the tropopause height affect the retrieved XCH₄, utsXCH₄, and troXCH₄. We will test this on a global scale by simulating an uncertainty of +/-33% of the climatological tropopause pressure (to pressures that are 33% lower/higher, which corresponds to a vertical up-/downward shift of about 3km). Moreover, we will document how a 10% increase of CH4 in the first layer above ground affects the retrieved XCH₄, utsXCH₄, and troXCH₄. We will add additional plots for JJA and DJF after Fig. 6.

Please note that in Schneider et al. (2022b), we already show the representativeness error, which is closely related to these tests and shows how typical CH4 variations are detected in the partial column products. There, we document a large representativeness error on the IASI troXCH₄ compared to the combined troXCH₄. This reveals that the IASI troXCH₄ is not really sensitive to surface-near CH4 and/or is strongly affected by CH4 in the upper troposphere / stratosphere, whereas the combined troXCH₄ mainly reflects tropospheric CH4.

4. Provide a workflow schematic (geo-match constraints in space/time/surface pressure; selection of the "best match"; then Kalman update; then layer integration). Summarize the role of the surface-pressure proximity filter in reducing representativeness error before the merge. (Readers will appreciate why Δpsfc and distance/time windows matter for the dislocation kernel.)

Yes, we agree: providing a workflow schematic summarizing the different processing steps will be helpful. We will add an additional figure with the schematic to the manuscript.

5. Interpret the covariance terms physically: when and why does dislocation error rise (latitudinal gradient of temporal mismatch; high terrain where the tro layer is shallow), and how this interacts with the first kilometer above ground (largest representativeness uncertainty). The text mentions these patterns; add a concise, quantitative example (e.g., Himalaya vs. adjacent lowlands).

Ok, we will try to improve the text in Section 5.2, providing a more detailed discussion of the observed dislocation error patterns.

6. Clarify how the DOFS patterns co-vary with the noise maps (you note this relationship; add a 2-D density scatter to quantify correlation). Also, discuss the implications for regional comparability across seasons/latitudes.

This anti-correlation between DOFS values and measurement noise is typical for optimal estimation methods, which minimize the a posteriori uncertainty. The a posteriori uncertainty $(S_{\hat{x}} = (S_a^{-1} + K^T S_{y,n}^{-1} K)^{-1}$, see Eq. (A5) in Schneider et al. 2022b) can also be written as the sum of the propagated a priori uncertainty (Eq. (A7) in Schneider et al., 2022b):

$$S_{\hat{x},r} = (A - I)S_a(A - I)^T = S_{\hat{x}} S_a^{-1} S_{\hat{x}} = \left(S_a^{-1} + K^T S_{y,n}^{-1} K\right)^{-1} S_a^{-1} \left(S_a^{-1} + K^T S_{y,n}^{-1} K\right)^{-1},$$

and the propagated measurement noise (Eq. (A6) in Schneider et al., 2022b):

$$S_{\hat{x},n} = G S_{y,n} G^T = S_{\hat{x}} K^T S_{y,n}^{-1} K S_{\hat{x}} = \left(S_a^{-1} + K^T S_{y,n}^{-1} K\right)^{-1} K^T S_{y,n}^{-1} K \left(S_a^{-1} + K^T S_{y,n}^{-1} K\right)^{-1}.$$

Here $A$ is the averaging kernel, $I$ the identity, $S_a$ the a priori uncertainty, $K$ the Jacobians (transformator from the measurement domain to the domain of the atmospheric states), and $G$ the gain matrix (for more details, please refer to Rodgers, 2000, or Schneider et al., 2022b, Appendix A). It is easy to prove that $S_{\hat{x}} = (S_a^{-1} + K^T S_{y,n}^{-1} K)^{-1} = S_{\hat{x},r} + S_{\hat{x},n}$. As can be seen from the two equations above, both $S_{\hat{x},r}$ and $S_{\hat{x},n}$ in-/decrease with in-/decreasing measurement noise ($S_{y,n}$) or in-/decreasing a priori uncertainty ($S_a$). De-/increasing $S_{\hat{x},r}$ and $S_{\hat{x},n}$ means de-/increasing measurement noise and in-/decreasing DOFS values, i.e., the anti-correlation as observed when comparing Figs. 5 and 7.

We will elaborate a plot that shows the covariance between DOFS and noise for XCH₄, utsXCH₄, and troXCH₄. This plot will then be shown in an additional Appendix to the manuscript.

7. Because troXCH₄ uncertainty grows over high terrain where the partial layer is shallow, add a topography-stratified analysis showing error growth and AK distortions with decreasing surface pressure, and recommend region-specific usage notes (e.g., Himalaya/Andes filters or uncertainty inflation).

We think that Fig. 6(c) is very clear in this context: it shows that for surface pressures below 750 hPa, the layer up to 450 hPa is too shallow to be fully detected ($K^T S_{y,n}^{-1} K$ representing the troposphere gets too small). The respective noise errors are then also increased (see also our reply to comment 6). We will elaborate on a plot that shows the noise error with respect to the surface pressure and the tropopause pressure (in line with the plot for DOFS, Fig. 6). This new plot will then be shown in an additional Appendix to the manuscript.

Nevertheless, please note that by separating the atmosphere at 50% surface pressure, the "artificial" troXCH₄ data quality degradation for high surface elevation will disappear. So, it is likely that for the revised data set, there will be no need for Fig. 6 and for a similar figure for the noise error (dependency of DOFS and noise on surface and tropopause pressure).

In this context, we can again refer to Table 2 (data filter recommendations).

8. A brief bias analysis for troXCH₄ XCH₄ against independent references (even if indirect) would greatly increase confidence.

We fully agree that the validation with independent reference data is very important for showing the reliability of a new data product. In this context, please note that this data set is extensively validated by detailed comparisons to data from TCCON, Aircore, and GAW (see Sect. 4 of Schneider et al., 2022b). We will better expose this existing validation work in the manuscript.